# CONTRASTIVE DIFFUSER: PLANNING TOWARDS HIGH RETURN STATES VIA CONTRASTIVE LEARNING

## ABSTRACT

Applying Diffusion in reinforcement learning for long-term planning has gained much attention recently. Depending on the capability of diffusion in modeling the underlying distribution, those methods leverage the diffusion to generate the subsequent trajectories for planning, and achieve significant improvement. However, these methods neglect the differences of samples in offline datasets, in which different states have different returns. They simply leverage diffusion to learn the distribution of data, and generate the trajectories whose states have the same distribution with the offline datasets. As a result, the probability of these models reaching the high-return states is largely dependent on the distribution in the dataset. Even equipped with the guidance model, the performance is still suppressed. To address these limitations, in this paper, we propose a novel method called CDiffuser, which devises a return contrast mechanism to pull the states in generated trajectories towards high-return states while pushing them away from low-return states. Experiments on 12 commonly used D4RL benchmarks demonstrate the effectiveness of our proposed method. Our code is publicly available at https://anonymous.4open.science/r/ContrastiveDiffuser.

## 1 INTRODUCTION

Offline reinforcement learning (offline RL) (Levine et al., 2020; Prudencio et al., 2023) has gained significant attention in recent years, where an agent is trained on pre-collected offline datasets and is evaluated online with the environment later. Since offline RL avoids potential risks from interacting with the environment during policy improvements, it has broad applications in numerous real-world scenarios, like commercial recommendation (Xiao & Wang, 2021), health care (Fatemi et al., 2022), dialog (Jaques et al., 2020) and autonomous driving (Shi et al., 2021).

While offline RL obviates costly online explorations, restricting the policy learning on static datasets poses additional challenges. Direct application of off-policy algorithms in offline scenarios comes with the extrapolation error problem (Fujimoto et al., 2019), which can cause inaccurate value estimations on out-of-distribution (OOD) actions to accumulate during Bellman backup. Extrapolation errors are alleviated in prior studies by adding conservative priors, *e.g.*, regularizing the policy (Fujimoto & Gu, 2021; Wu et al., 2019) or penalizing the value estimations (Kumar et al., 2020; Kostrikov et al., 2021). However, such conservative updates may leave the learning policy trapped in local optima, especially when offline datasets are collected by a mixture of policies (Wang et al., 2022). Recently, diffusion models have been used in offline RL as a powerful policy class (Pearce et al., 2023; Chi et al., 2023; Ada et al., 2023; Wang et al., 2022). Due to diffusion models' ability to model arbitrary distributions, using them to fit the entire dataset can effectively regularize the policy without concerning of lacking expressiveness.

Among diffusion-based offline RL methods, a common approach is to utilize diffusion for long-term planning (Ajay et al., 2023; Janner et al., 2022). Specifically, these methods leverage the diffusion model to generate subsequent trajectories, which include state-action pairs in a period of future. The generated trajectories carry the estimated future states and enrich the information for planning, therefore they enhance models to make better decisions to be taken in the environment. However, these methods neglect the diversity of samples in offline datasets, in which different states have different returns. They simply leverage diffusion to learn the dataset distribution and generate the trajectories whose states share the same distribution with the offline dataset. As shown in Figure 1, the state

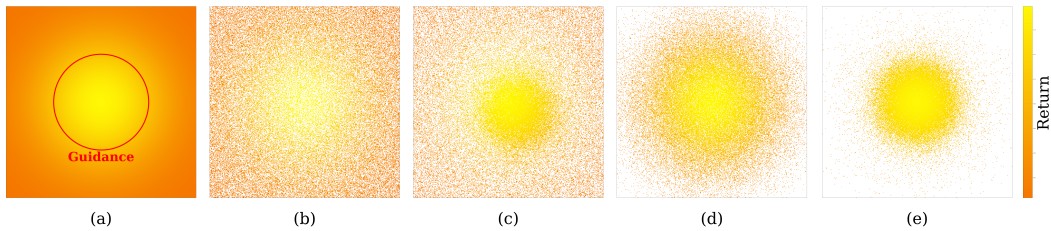

Figure 1: Comparison of different distributions: (a) The dataset distribution; (b) The uniform sampling of diffusion; (c) The classifier guide sampling of diffusion; (d) The improved sampling of diffusion; (e) The improved guidance sampling of distribution. Each scatter in sub-figure represents a two-dimensional state, and the color of each scatter denotes the corresponding return.

distribution learned by diffusion (b) is similar to the original distribution of the offline dataset (a), which makes the probability of sampling high-return states relatively low if there are many low-return states in the dataset. Even with guided sampling (Dhariwal & Nichol, 2021) techniques to enforce the generation towards the high-return region, the results remain unsatisfactory as depicted in Figure 1(c). Although the sampling distribution is more concentrated within the guidance circle, there are still many samples spread over the entire low-return part. Intuitively, if constraining the trajectory generated by diffusion to close to the area with high-return states and away from the area with low-return states, like Figure 1 (d), we would obtain better results under the guidance, like Figure 1(e).

Considering contrastive learning (CL) (Khosla et al., 2020; Yeh et al., 2022) is designed for pulling a sample towards the similar samples and pushing it away from dissimilar samples, which is analogous to the case of pulling the states in the generated trajectory towards the high-return areas and away from low-return areas, we propose a novel method called **Contrastive Diffuser** (**CDiffuser**). Different from the previous works (Qiu et al., 2022; Laskin et al., 2020; Yuan & Lu, 2022; Agarwal et al., 2020) which leverage CL for the representation learning in RL, we introduce CL to bias the diffusion model training with return contrasting. Specifically, we group the states in the offline dataset into high-return states and low-return states in a soft manner. Then, we learn a diffusion-based trajectory generation model to generate the trajectories whose states are constrained by contrastive learning to keep close to the high-return states and away from the low-return states. With the help of contrastive learning, CDiffuser generates better trajectories for planning. To evaluate the performance of CDiffuser, we conduct experiment on 12 D4RL (Fu et al., 2020) benchmarks. The experiment results demonstrate that CDiffuser has superior performance.

In summary, our contributions are as follows: (i) We propose a novel method called CDiffuser, which improves the performance of diffusion based RL algorithms. (ii) We perform contrastive learning over returns of states. To the best of our knowledge, our work is the first which apply contrastive learning to contrast the return to enhance the diffusion model training in RL. (iii) Experiment results on D4RL datasets demonstrate the outstanding performance of CDiffuser.

## 2 BACKGROUND

### 2.1 DENOISING PROBABILISTIC MODELS

Denoising Probabilistic Models (Diffusion Models) (Sohl-Dickstein et al., 2015; Song et al.; Ho et al., 2020) are a group of generative models, which generate samples by denoising from Gaussian noises. A diffusion model is composed of a forward process and a backward process. Given the original data $\boldsymbol{x}^0 \sim q(\boldsymbol{x}^0)$, the forward process transfers $\boldsymbol{x}^0$ into a Gaussian noise by gradually adding noises, i.e., $q(\boldsymbol{x}^i|\boldsymbol{x}^{i-1}) = \mathcal{N}(\boldsymbol{x}^i; \sqrt{1-\beta^i}\boldsymbol{x}^{i-1}, \beta^i\boldsymbol{I})$, in which $\boldsymbol{I}$ is an identity matrix, $\beta^i$ is the noise schedule measuring the proportion of noise added at each step. The reverse process recovers $\boldsymbol{x}^0$ by gradually removing the noise and each step, which is formulated with a Gaussian distribution (Feller, 1949) parameterized by $\theta$, i.e., $p_\theta(\boldsymbol{x}^{i-1}|\boldsymbol{x}^i) = \mathcal{N}(\mu_\theta(\boldsymbol{x}^i, i), \Sigma_\theta(\boldsymbol{x}^i, i)), \bar{\alpha}^i = \prod_{j=1}^i (1-\beta^i)$.

Following DDPM (Ho et al., 2020), the objective function can be formulated as follows if we fix $\boldsymbol{\Sigma}_\theta(\boldsymbol{x}_t, t) = \beta_t \boldsymbol{I}$:

$$\mathcal{L} = \mathbb{E}_{\boldsymbol{x}^0, \, i \sim [1,N]}[\|\boldsymbol{x}^0 - \psi_\theta(\boldsymbol{x}^i, i)\|^2], \tag{1}$$

where $\psi_\theta(\cdot, \cdot)$ is a model to reconstruct $\boldsymbol{x}^0$.

## 2.2 CONTRASTIVE LEARNING

Contrastive learning (Schroff et al., 2015; Sohn, 2016; Khosla et al., 2020; Yeh et al., 2022; Oord et al., 2018) is a class of self-supervised learning method which aims at pulling similar samples together and pushing different samples away. Specifically, given a sample $x$ and a similarity measure, the positive sample $x^+$ is defined as the sample similar to $x$, and the negative set $\mathcal{S}^-$ is defined as the collection of samples dissimilar to $x$. Contrastive learning minimizes the distance of between $x$ and $x^+$, and maximizes the distance between $x$ and $\mathcal{S}^-$. That is:

$$\mathcal{L} = -\log \left[ \frac{\exp(\text{sim}(f(x), f(x^+)))}{\exp(\text{sim}(f(x), f(x^+))) + \sum_{x^- \in \mathcal{S}^-} \exp(\text{sim}(f(x), f(x^-)))} \right], \tag{2}$$

where $f(\cdot)$ denotes the function that mapping samples to a latent space and $\text{sim}(\cdot, \cdot)$ denotes the similarity measure.

## 2.3 PROBLEM SETTING

Considering a system composed of three parts: policy, agent, and environment. The environment in RL is usually formulated as a Markov Decision Process (MDP) (Sutton & Barto, 2018) $\mathcal{M} = \{\mathcal{S}, \mathcal{A}, \mathcal{P}, r, \gamma\}$, where $\mathcal{S}$ is the state space, $\mathcal{A}$ is the action space, $\mathcal{P}(s'|s, a)$ is the transition function, $\gamma$ represents the discount factor, $r$ is the instant reward of each step. At each step $t$, the agent respond to the state of environment $s_t$ by action $a_t$ according to policy $\pi_\theta$ parameterized by $\theta$, and gets an instant return $r_t$. The interaction history is formulated as a trajectory $\tau = \{(s_t, a_t, r_t)|t \geq 0\}$. Please notice that in this paper, we define the cumulative discounted reward from step $t$ as $v_t = \sum_{i \geq t} \gamma^{i-t} r_i$ and call it as the return of $s_t$.

We focus on the offline RL setting in this paper. Therefore, given an offline dataset $\mathcal{D} \triangleq \{(s_t, a_t, r_t, s_{t+1})|t \geq 0\}$ consisting of transition tuples, and defining the return of trajectory $\tau$ as $R(\tau) \triangleq \sum_{t \geq 0} \gamma^t r_t$, our goal is learning $\pi_\theta$ to maximize the expected return without directly interacting with the environment, *i.e.*,

$$\pi_\theta = \arg\max_\theta \mathbb{E}_{\tau \sim \pi_\theta}[R(\tau)] . \tag{3}$$

## 3 METHODOLOGY

Following Diffuser (Janner et al., 2022), we formulate the offline RL problem as a state-conditioned sequence generative task. To tackle the limitation of overlooking sample differences in prior works, we propose a method called CDiffuser, which introduces contrastive learning and addresses the limitation with a return contrast mechanism. Specifically, our CDiffuser is composed of two modules: (1) the Planning Module, which aims to generate subsequent trajectories; (2) the Contrastive Module, which is designed to keep the states in generated trajectories within the high-return regions but away from low-return states, as is illustrated in Figure 2.

## 3.1 PLANNING MODULE

Following Diffuser (Janner et al., 2022), given a state $s_t$ at step $t$, the Planning Module estimates $v_t$ as guidance, and leverages the guidance as well as $s_t$ as the condition to generate the subsequent trajectory, as is illustrated in Figure 2. Specifically, we first sample $\hat{\tau}_t^N$ from $\mathcal{N}(\mathbf{0}, \mathbf{I})$, and replace $\hat{s}_t^N$ with $s_t$ as condition on the current observation:

$$\hat{\tau}_t^N = \{(s_t, \hat{a}_t^N), (\hat{s}_{t+1}^N, \hat{a}_{t+1}^N), ..., (\hat{s}_{t+H}^N, \hat{a}_{t+H}^N)\} , \tag{4}$$

in which all the elements except $s_t$ are pure Gaussian noise. We further feed $\hat{\tau}_t^N$ into the reverse process to generate the subsequent trajectory:

$$p_\theta(\hat{\tau}_t^{i-1}|\hat{\tau}_t^i) = \mathcal{N}(\mu_\theta(\hat{\tau}_t^i, i) + \rho \nabla \mathcal{J}_\phi(\hat{\tau}_t^i, i), \beta_i \mathbf{I}) , \tag{5}$$

$$\mu_\theta(\hat{\tau}_t^i, i) = \frac{\sqrt{\alpha^i}(1 - \bar{\alpha}^{i-1})}{1 - \bar{\alpha}^{i-1}} \hat{\tau}_t^i + \frac{\sqrt{\bar{\alpha}^{i-1}}\beta^i}{1 - \bar{\alpha}^i} \hat{\tau}_t^{i,0} . \tag{6}$$

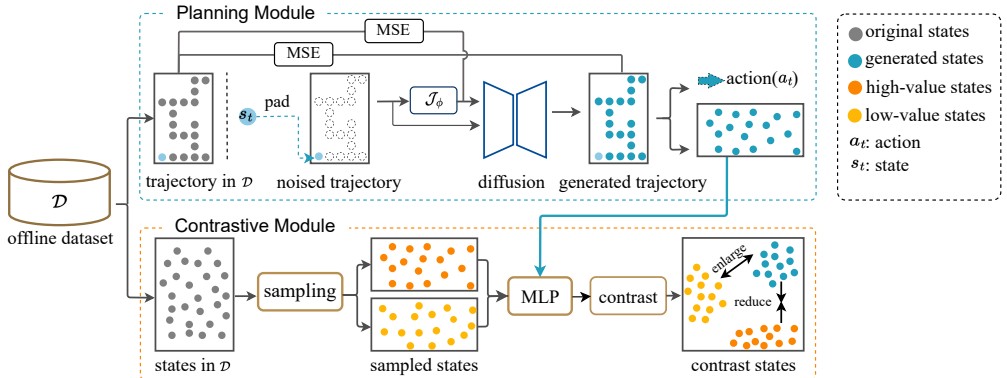

Figure 2: The overall framework of CDiffuser. CDiffuser is composed of two modules, namely the the Planning Module and the Contrastive Module. The Planning Module is designed to generate the subsequent trajectories, and the Contrastive Module is designed to pull the states in the generated trajectories toward the high-return states and push them away from the low-return states during the training phase.

Here $\hat{\tau}_t^{i,0} = \psi_\theta(\hat{\tau}_t^i, i)$ represents the $\tau_t^0$ constructed from $\hat{\tau}_t^i$ at diffusion step $i$, $\psi_\theta(\cdot, \cdot)$ is a network for trajectory generation, $i \sim [1, N]$ is the diffusion step, $\rho$ represents the guidance scale, $\mathcal{J}_\phi(\cdot, \cdot)$ is a learned function to predict the return given any noisy trajectory $\tau_t^i$. We abbreviate $\hat{\tau}_t^0$ to $\hat{\tau}_t$ for convenience, and denote it as $\hat{\tau}_t = \{(s_t, \hat{a}_t), (\hat{s}_{t+1}, \hat{a}_{t+1}), ..., (\hat{s}_{t+H}, \hat{a}_{t+H})\}$. We take out the $\hat{a}_t$ in $\hat{\tau}$ as the action corresponding to the state $s_t$.

## 3.2 CONTRASTIVE MODULE

Although the Planning Module can independently generate the action responding to the environment, its performance is limited due to neglecting the differences of training samples. Fortunately, this can be improved via the Contrastive Module, which adopts contrastive learning to pull the planned states toward the high-return states and push them away from the low-return states. Note that different from the previous works (Laskin et al., 2020; Qiu et al., 2022; Yuan & Lu, 2022; Agarwal et al., 2020) which apply contrastive learning to obtain a

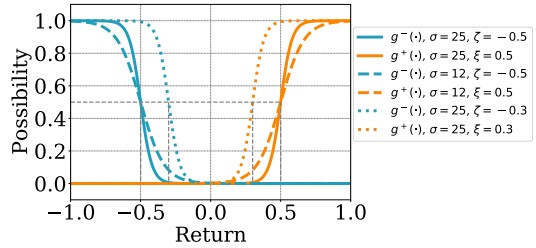

Figure 3: Modified influence functions.

better representation, we contrast the return of states for reaching high-return states. In the following parts of this section, we first introduce the construction of contrastive sample sets, and then we explain how we perform contrastive learning to improve the trajectory generation in the Planning Module.

### 3.2.1 SAMPLE POSITIVE AND NEGATIVE STATES

The positive samples and negative samples are necessary before applying contrastive learning. Intuitively, we can naively use hard thresholds to split states into positive and negative sets. However, such a radical method is unable to fully utilize samples located near the boundaries. Thus, we propose to conduct probabilistic partitioning.

Specifically, for an arbitrary state $s_t$ in the offline dataset, we compute its return $v_t$ first. Then, we adopt modified influence functions (Thoma et al., 2020), $g^+(\cdot)$ and $g^-(\cdot)$, to perform soft classification, determining the probability of classifying $s_t$ as a positive sample or negative sample:

$$p^+(s_t) \triangleq g^+(v_t) = \frac{1}{1 + e^{\sigma(\xi - v_t)}}, \tag{7}$$

$$p^-(s_t) \triangleq g^-(v_t) = \frac{1}{1 + e^{\sigma(v_t - \zeta)}}, \tag{8}$$

where $p^+(s_t)$ denotes the probability of $s_t$ being grouped into positive samples, and $p^-(s_t)$ denotes the probability of $s_t$ being grouped into negative samples.

In our modified influence functions, $\xi$ and $\zeta$ are the fuzzy centers of boundaries of positive and negative samples, $\sigma$ represents the fuzzy coefficient. As is shown in Figure 3, with $\xi$ getting larger, fewer samples are grouped into positive samples; With $\zeta$ getting smaller, fewer samples are grouped into negative samples; A larger $\sigma$ makes $g^+(v_t)$ and $g^-(v_t)$ sharper.

### 3.2.2 CONSTRAIN THE TRAJECTORY WITH CONTRASTIVE LEARNING

Following Kang et al. (2023), instead of running the whole reverse denoising process to sample $\hat{\tau}_t$ for contrastive, we cheaply contruct $\hat{\boldsymbol{\tau}}_t^{i,0} = \{(\hat{\boldsymbol{s}}_t^{i,0}, \hat{\boldsymbol{a}}_t^{i,0}), (\hat{\boldsymbol{s}}_{t+1}^{i,0}, \hat{\boldsymbol{a}}_{t+1}^{i,0}), ..., (\hat{\boldsymbol{s}}_{t+H}^{i,0}, \hat{\boldsymbol{a}}_{t+H}^{i,0})\}$ from $\boldsymbol{\tau}_t^i$ by performing one-step denoising.

To constrain the states in this trajectory, we extract states in $\hat{\boldsymbol{\tau}}_t^{i,0}$ as $\mathcal{S}_{\hat{\boldsymbol{\tau}}_t^{i,0}} = \{\hat{\boldsymbol{s}}_{t+1}^{i,0}, \hat{\boldsymbol{s}}_{t+2}^{i,0}, ..., \hat{\boldsymbol{s}}_{t+H}^{i,0}\}$ first. Next, for each state $\hat{\boldsymbol{s}}_h^{i,0} \in \mathcal{S}_{\hat{\boldsymbol{\tau}}_t^{i,0}}$, we sample $\kappa$ states via Equation (7) as the positive samples and sample $\kappa$ states via Equation (8) as negative samples, denoted as $\mathcal{S}_h^+$ and $\mathcal{S}_h^-$ correspondingly. Inspired by Schroff et al. (2015); Sohn (2016), we adopt the following equation to pull the states in the generated trajectory toward the high-return states and away from the low-return states:

$$\mathcal{L}_h^i = -\log \frac{\sum_{k=0}^{\kappa} \exp(\text{sim}(f(\hat{\boldsymbol{s}}_h^{i,0}), f(\boldsymbol{s}_h^+))/T)}{\sum_{k=0}^{\kappa} \exp(\text{sim}(f(\hat{\boldsymbol{s}}_h^{i,0}), f(\boldsymbol{s}_h^-))/T)} , \qquad (9)$$

where $\boldsymbol{s}_h^+ \in \mathcal{S}_h^+$, $\boldsymbol{s}_h^- \in \mathcal{S}_h^-$, $f(\cdot)$ represents the projection function, $T$ represents the temperature, and $\text{sim}(\cdot, \cdot)$ denotes the similarity measure, which is computed as

$$\text{sim}(\boldsymbol{a}, \boldsymbol{b}) = \frac{\boldsymbol{a}^\top \boldsymbol{b}}{\|\boldsymbol{a}\| \cdot \|\boldsymbol{b}\|} . \qquad (10)$$

### 3.3 MODEL LEARNING

Recall that the action responding to state $\boldsymbol{s}_t$ is one of the elements in the generated trajectory, and is influenced by $\mathcal{J}_\phi(\cdot, \cdot)$ and contrastive learning. Therefore, we optimize our method from the perspective of trajectory generation, return prediction and trajectory generation constrain.

Specifically, we optimize the trajectory generation by minimizing the Mean Square Error between the ground truth and clean trajectory predicted by $\psi_\theta(\cdot, \cdot)$ given any intermediate noisy trajectories as input:

$$\mathcal{L}_d = \mathbb{E}_{\boldsymbol{\tau}_t \in \mathcal{D}, t>0, i \sim [1,N]} \left[ \|\boldsymbol{\tau}_t - \psi_\theta(\boldsymbol{\tau}_t^i, i)\|^2 \right] , \qquad (11)$$

where $i$ denotes the step of diffusion, $\boldsymbol{\tau}_t^i$ is obtained in the $i$-th step of forward process. We optimize the return prediction by minimizing the Mean Square Error between the predicted return $\mathcal{J}_\phi(\boldsymbol{\tau}_t^i, i)$ and the ground-truth return $v_t$:

$$\mathcal{L}_v = \mathbb{E}_{\boldsymbol{\tau}_t \in \mathcal{D}, t>0, i \sim [1,N]} [\|\mathcal{J}_\phi(\boldsymbol{\tau}_t^i, i) - v_t\|^2] . \qquad (12)$$

We constrain the trajectory generation with a reweighted contrastive loss:

$$\mathcal{L}_c = \mathbb{E}_{t>0, i \sim [1,N]} \left[ \sum_{h=t}^{t+H} \frac{1}{h+1} \mathcal{L}_h^i \right], \qquad (13)$$

in which the coefficient $\frac{1}{h+1}$ decreases as $h$ increases since the impact of predictions in the future on planning is smaller.

Hence, the overall objective function of CDiffuser can be written as a weighted sum of the aforementioned loss terms:

$$\mathcal{L} = \lambda_d \mathcal{L}_d + \lambda_v \mathcal{L}_v + \lambda_c \mathcal{L}_c , \qquad (14)$$

where $\lambda_d$, $\lambda_v$, $\lambda_c$ are hyperparameters, which balance the importances of the corresponding learning targets. Please notice that optimizing the return predictor $\mathcal{J}\phi(\cdot, \cdot)$ with Equation (14) is equal to optimizing it with Equation (12) only, we put the objectives together in Equation (14) for neatness. Please refer to Appendix A.5 for details.

The pseudo code of CDiffuser is presented in Appendix A.1, and the detail of implementation will be discussed in the next section.

## 4 EXPERIMENTS

In this section, we evaluate the performance of CDiffuser in three locomotion environments under three settings, as well as a navigation environment under three settings.

### 4.1 EXPERIMENT SETTINGS

**Environments and datasets.** Following Diffuser (Janner et al., 2022), we evaluate the performance of CDiffuser on the locomotion tasks and navigation tasks. Specifically, we evaluate the locomotion capability of CDiffuser on the environment of Halfcheetah, Hopper, Walker2d, and we evaluate the navigation capability of CDiffuser on the environment of Maze2d. For each environment, we train CDiffuser with three scales of offline datasets provided by D4RL (Fu et al., 2020), and test the performance of CDiffuser on the corresponding environment.

**Baselines.** We compare CDiffuser with diffusion-free methods such as CQL (Kumar et al., 2020), IQL (Kostrikov et al., 2021), MOPO (Yu et al., 2020), Decision Transformer (DT) (Chen et al., 2021) and Trajectory Transformer (TT) (Janner et al., 2021). Further, we compare CDiffuser with diffusion-based methods, such as Diffuser (Janner et al., 2022) and Decision Diffuser (DD) (Ajay et al., 2023), which apply diffusion to model RL as sequence generation problems.

**Implementation details.** We adopt U-Net (Ronneberger et al., 2015) as the denoise network $\psi_\theta(\cdot, \cdot)$ and the return predictor $\mathcal{J}_\phi(\cdot, \cdot)$, and adopt a linear layer with $Sigmoid$ as the activation function as the projector $f(\cdot)$. Our model is trained on a device with 4 NVIDIA A40 GPUs (48GB GPU memory, 37.4 TFLOPS computing capabilities), Intel Gold 5220 CPU (72 cores, 2.20GHz clock frequency) and 504G memory, optimized by Adam (Kingma & Ba, 2014) optimizer. Details of hyper-parameters are shown in Appendix A.3.

Table 1: The average normalized score of different methods on various environments, with $\pm$ denoting the variance. The mean and std are computed over 10 random seeds. The best and the second-best results of each setting are marked as **bold** and underline, respectively.

| Dataset | Environment | CQL | IQL | DT | TT | MOPO | Diffuser | DD | CDiffuser |
|---------|-------------|-----|-----|-----|-----|------|----------|-----|-----------|
| Med-Expert | HalfCheetah | 91.6 | 86.7 | 86.8 | **95.0** | 63.3 | 88.9 | 90.6 | 92.0 $\pm$ 0.4 |
| Med-Expert | Hopper | 105.4 | 91.5 | 107.6 | 110.0 | 23.7 | 103.3 | 111.8 | **112.4** $\pm$ **1.2** |
| Med-Expert | Walker2d | 108.8 | **109.6** | 108.1 | 101.9 | 44.6 | 106.9 | 108.8 | 108.2 $\pm$ 0.4 |
| Medium | HalfCheetah | 44.0 | 47.4 | 42.6 | 46.9 | 42.3 | 42.8 | **49.1** | 43.9 $\pm$ 0.9 |
| Medium | Hopper | 58.5 | 66.3 | 67.6 | 61.1 | 28.0 | 74.3 | 79.3 | **92.3** $\pm$ **2.6** |
| Medium | Walker2d | 72.5 | 78.3 | 74.0 | 79.0 | 17.8 | 79.6 | 82.5 | **82.9** $\pm$ **0.5** |
| Med-Replay | HalfCheetah | 45.5 | 44.2 | 36.6 | 41.9 | **53.1** | 37.7 | 39.3 | 40.0 $\pm$ 1.1 |
| Med-Replay | Hopper | 95 | 94.7 | 82.7 | 91.5 | 67.5 | 93.6 | **100** | 96.4 $\pm$ 1.1 |
| Med-Replay | Walker2d | 77.2 | 73.9 | 66.6 | 82.6 | 39.0 | 70.6 | 75 | **84.2** $\pm$ **1.2** |
| U-Maze | Maze2d | 5.7 | 47.4 | - | - | - | 113.9 | - | **142.9** $\pm$ **2.2** |
| Medium | Maze2d | 5.0 | 34.9 | - | - | - | 121.5 | - | **140.0** $\pm$ **0.7** |
| Large | Maze2d | 12.5 | 58.6 | - | - | - | 123.0 | - | **131.5** $\pm$ **3.2** |

### 4.2 BENCHMARK RESULTS

We compare CDiffuser to baseline methods with respect to the normalized average returns (Fu et al., 2020) obtained during online evaluation. We conducted 10 trials with different seeds and reported the average results. The results of CDiffuser and baseline methods are summarized in Table 1.

From Table 1, we can observe that: (1) Compared with all the baseline methods, CDiffuser achieves the best or the second-best performance on 6 out of 9 locomotion tasks, demonstrating the outstanding performance of CDiffuser under periodic settings. Moreover, CDiffuser achieves the best performance on all the three navigation tasks, demonstrating the excellent ability of CDiffuser in long-term planning. (2) Compared with our backbone method Diffuser, CDiffuser outperforms Diffuser in all the 12 tasks, which demonstrates the effectiveness of contrast in boosting diffusion-based RL methods. Moreover, CDiffuser exhibits more improvement in medium and medium-replay datasets than

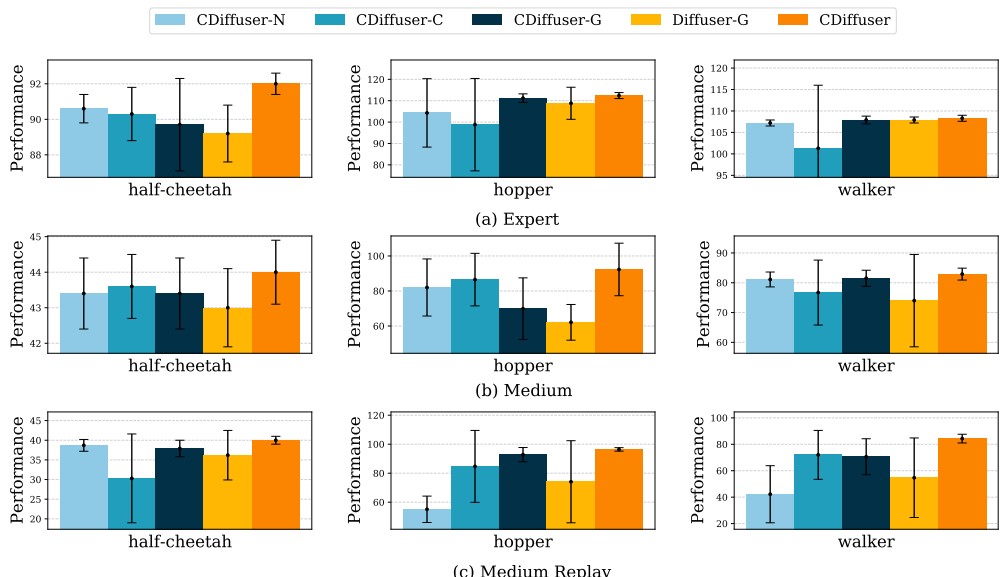

Figure 4: Results of the ablation experiments on different variants.

the expert dataset. We believe that is because the expert datasets have more high-return samples, which makes it easier for Diffuser to learn and achieve better results. However, both medium and medium-replay have more low-return samples, which increases the difficulty for Diffuser to learn a good policy. These results demonstrate that CDiffuser is better at making use of low-return samples.

### 4.3 ABLATION STUDIES

We conduct abalation studies to further investigate the impact of contrasting returns on performance. Specifically, we explore the following four variants:

- **CDiffuser-C**: remove contrastive learning from CDiffuser, *i.e.*, remove $\mathcal{L}_c$ from Equation (14).
- **CDiffuser-N**: only apply the samples with high-return to train the model.
- **CDiffuser-G**: remove the guidance from CDiffuser, *i.e.*, removing $\rho \nabla \mathcal{J}_\phi(\cdot, \cdot)$ from Equation (5).
- **Diffuser-G**: remove the classifier guidance from Diffuser.

The results are summarized in Figure 4. From Figure 4, we can observe that: (1) CDiffuser surpasses CDiffuser-C, illustrating the clear benefits of contrasting the trajectory generation process with high-return and low-return samples; (2) CDiffuser-G outperforms Diffuser-G in 8 out of 9 datasets. Since the only difference is whether using the contrastive learning, the result demonstrates contrasting with high-return and low-return samples is effective in improving online performance; (3) CDiffuser-N underperforms CDiffuser in all the cases. Since CDiffuser-N applies no negative samples, this phenomenon demonstrates the success of performing contrastive learning with both positive and negative samples. (4) CDiffuser-N underperforms CDiffuser-C in 4 out of 9 cases. We argue that since CDiffuser-N is trained using only a small portion of samples (*i.e.*, positive samples), this results in its inability to learn information from the discarded samples, leading to worse performance than CDiffuser-C, which is trained over the whole dataset; (5) CDiffuser-G is better than CDiffuser-C in most cases, especially in medium and medium-replay. That implies the constraint of states' return is more useful than the guidance in the cases like medium or medium-replay, in which the numbers of high-return samples are limited.

### 4.4 FURTHER INVESTIGATION

To further investigate the performance of CDiffuser, we analyze the state-reward distribution and the long-term dynamic consistency.

**State-reward distribution analysis.** we randomly collect the (state, reward) pairs from the offline dataset of Walker2d-Med-Replay and the (state, reward) pairs collected when Diffuser, Decision Diffuser, and CDiffuser interact with the environment, and compare them in Figure 5. Here, we choose Diffuser and Decision Diffuser as both of them apply diffusion to model RL as a sequence

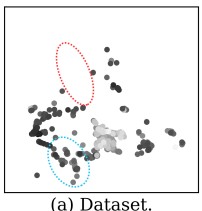 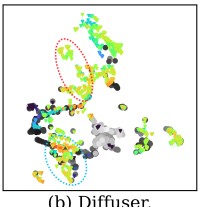 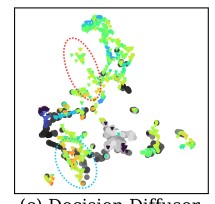 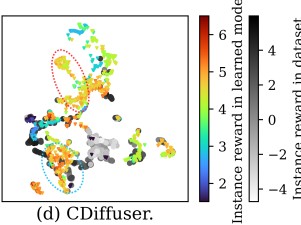

| (a) Dataset. | (b) Diffuser. | (c) Decision Diffuser. | (d) CDiffuser. |

Figure 5: The distribution of state and reward. It is better to view in color mode. CDiffuser achieves higher rewards in both in-distribution areas(circled with blue) and out-of-distribution areas(circled with red).

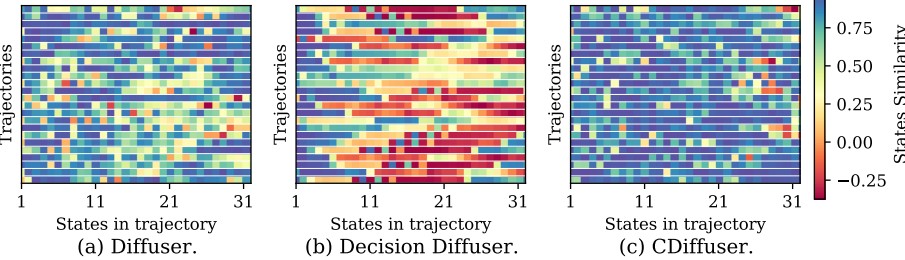

(a) Diffuser.     (b) Decision Diffuser.     (c) CDiffuser.

Figure 6: The similarities between the states in the generated trajectories and actual states. The generated states of CDiffuser are more similar with the actual states, demonstrating the better long-term dynamic consistency.

generation problem. In Figure 5, each scatter represents a state, and its color denotes the reward grained in the corresponding state. From the results illustrated in Figure 5, we can observe that: in both in-distribution states(circled with blue) and out-of-distribution states(circled with red), our CDiffuser gains higher rewards. We suppose that because the contrastive module enhances CDiffuser's long-term dynamic consistency, which represents the similarity of the states in the generated trajectories and the actual states provided by environment. According to Equation (5) and Equation (6), the long-term dynamic consistency benefits the decision making of CDiffuser.

**Long-term dynamic consistency analysis.** To further investigate whether the contrastive module enhances the long-term dynamic consistency of CDiffuser, we randomly take 24 trajectories generated by Diffuser, Decision Diffuser, and CDiffuser. For each generated trajectory, we take the states of consecutive 32 steps and compute the similarity between each generated state and the actual state of the same step provided by the environment. Thus, there are $24 \times 32$ similarity values for each model, which corresponds to a similarity matrix as the subgraphs in Figure 6 illustrated. Each line in the subgraphs of Figure 6 represents a generated trajectory, and the grids of each line represent the similarity of the states in the generated trajectory and the states provided by the environment. From Figure 6, we can observe that: (1) Most grids in Figure 6 (c) are blue, which denotes that most generated states consistent with the actual states; (2) Figure 6 (c) contains more blue grids than Figure 6 (a) and (b), which denotes that CDiffuser has better long-term dynamic consistency than Diffuser and Decision Diffuser. Since the difference between CDiffuser and Diffuser is the contrastive module, combining Figure 5 and Figure 6, we can conclude that the contrative module benefits the long-term dynamic consistency of CDiffuser, making it gain high rewards in both in-distribution and out-of-distribution situations.

### 4.5 HYPERPARAMETER ANALYSIS

We conduct additional experiments to investigate the impact of different hyper-parameters on the performance. Specifically, we evaluate the performance of CDiffuser under different $\xi$, $\zeta$, $\sigma$ and $\lambda_c$. In these experiments, all the settings remain the same except the value of the tested hyper-parameter. The experiment results are illustrated in Figure 7.

In the result presented in Figure 7, we can find: (1) with the increase of $\xi$ (Figure 7(a)), the performance gradually increases but the decreases when $\xi > 0.85$. The underlying reason is that with $\xi$ increases, the proportion of high-return states in positive samples increases, leading to an improvement in model performance. However, as $\xi$ gradually becomes larger, the available samples for contrasting decreases, resulting in a decline in performance. We can observe a similar pattern with $\zeta$ decreases, as is shown in Figure 7(b). (2) With the increase of $\sigma$, the performance gradually

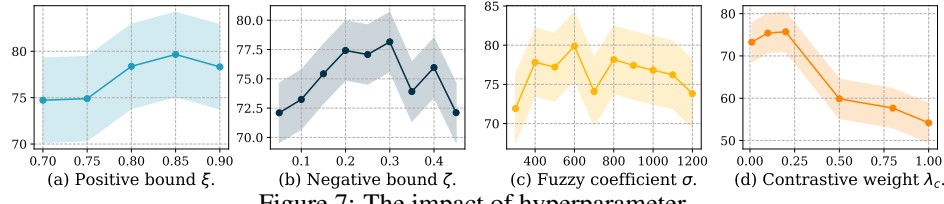

Figure 7: The impact of hyperparameter.

increases and then decreases. We argue that since it is difficult to confidently classify samples near the boundary as positive or negative, an appropriate $\sigma$ provides efficient tolerance for the classification of these samples. However, a low $\sigma$ blurs the boundary between positive and negative samples, while a high $\sigma$ loses the aforementioned tolerance, thus resulting in worse performance. (3) With the increases of $\lambda_c$ , the performance increases at the very steps but decreases then. We conclude that increasing the weight of contrasting leads the generated states towards high-return states. However, over-emphasizing the contrast will lead to neglecting dataset distribution, thus losing the generalization of diffusion and resulting in a decrease in performance. (4) It can be observed that CDiffuser exhibits a smooth and regular change in performance with hyperparameters various, which makes it easier for us to tune the parameters.

## 5 RELATED WORKS

### 5.1 DIFFUSION IN DECISION MAKING

We group the diffusion-based methods in RL into action generation methods and trajectory generation methods. The action generation methods (Ada et al., 2023; Wang et al., 2022; Chen et al., 2022; Chi et al., 2023) adopt diffusion models as policies to predict the action of the current step. One of the typical works in this group is Diffusion Q-learning (Wang et al., 2022), which proposes to design policy as a diffusion model and improve it with double Q-learning architecture. Following Diffusion Q-Learning, SRDPs (Ada et al., 2023) incorporates state reconstruction feature learning into the recent category of diffusion policies to address the out-of-distribution generalization problem. The second group of methods generate the subsequent trajectory including the action to take at the current step by diffusion. For instance, Diffuser (Janner et al., 2022) models trajectories as sequences of state-action pairs. Based on Diffuser, Decision Diffuser (Ajay et al., 2022) proposes to predict state sequences with a diffusion model conditioned on historical information, and adopts a reverse dynamic model to predict actions based on the generated state sequence. Though these methods have gain significant achievements, they neglect the differences in samples.

### 5.2 CONTRASTIVE LEARNING IN RL

The motivation for introducing contrastive learning in RL is to enrich the representation in the previous works. We group these works into three types. The first type of methods apply contrastive learning to enhance the state representations (Laskin et al., 2020; Qiu et al., 2022). For instance, Laskin et al. (2020) propose to learn image representations via contrastive learning; Qiu et al. (2022) propose to learn the transition with contrastive learning. The second type of methods apply contrastive learning to learn the representations of tasks. For instance, Yuan & Lu (2022) apply contrastive learning to enhance the representation of tuples to distinguish between different tasks; Agarwal et al. (2020) apply contrastive learning to learn the representations of the environments. Some works apply contrastive learning in other ways. For instance, Laskin et al. (2022) utilizes contrastive learning to learn behavior representations and maximizes the entropy to encourage behavioral diversity. In contrast to the methods mentioned above, CDiffuser adopts contrastive learning to constrain the generated sample, rather than learning representations.

## 6 CONCLUSION AND DISCUSSION

In this paper, we introduce CDiffuser for offline RL, which introduces contrastive learning to constrain the trajectory generation. Different from the previous works which apply contrastive learning to enhance the representation, we contrast the return of states. Specifically, we apply diffusion to generate the subsequent trajectory for planning, and then we constrain the states in the generated trajectory toward the states with high returns and away from the states with low returns. In that way, the actions taken by the agent are always toward the high-return states, which makes the agent gain better performance in the online evaluation. We evaluated CDiffuser on 12 D4RL benchmarks, the results demonstrate that our CDiffuser achieves outstanding performance. However, the CDiffuser is limited in the case in which a certain state corresponds with both high and low return, as CDiffuser relies the return on the state to contrast. Nevertheless, the contrast based on the return of state is just the beginning, the contrast on actions also deserves to be explored. We will leave it to future works.

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

# A  APPENDIX

## A.1  PSEUDOCODE OF CDIFFUSER.

---
**Algorithm 1** Training
---
1: Calculate the candidate set $\mathcal{C}$.
2: **while** not converged **do**
3:     $\boldsymbol{\tau}_t, v_t \sim \mathcal{D}$.
4:     $i \sim [1, N]$.
5:     Generate $\boldsymbol{\tau}_t^i$.
6:     Reconstruct $\boldsymbol{\tau}_t$ as $\hat{\boldsymbol{\tau}}_t^{i,0} = \psi_\theta(\boldsymbol{\tau}_t^i, i)$.
7:     Calculate loss $\mathcal{L}_d$ with Equation (11).
8:     Calculate loss $\mathcal{L}_v$ with Equation (12).
9:     Extract states in $\hat{\boldsymbol{\tau}}_t^{i,0}$ as $\mathcal{S}_{\hat{\boldsymbol{\tau}}_t^{i,0}} = \{\hat{\boldsymbol{s}}_{t+1}^{i,0}, \hat{\boldsymbol{s}}_{t+2}^{i,0}, ..., \hat{\boldsymbol{s}}_{t+H}^{i,0}\}$.
10:     **for** $\hat{s}_h^{i,0}$ in $\mathcal{S}_{\hat{\boldsymbol{\tau}}_t^{i,0}}$ **do**
11:         Sample $\mathcal{S}^+$ and $\mathcal{S}^-$ with Equation (7) and Equation (8).
12:         Calculate $\mathcal{L}_h^i$ using Equation (9).
13:     **end for**
14:     Calculate $\mathcal{L}_c$ using Equation (13).
15:     Calculate $\mathcal{L}$ using Equation (14).
16:     Update model by taking gradient decent with $\mathcal{L}$.
17: **end while**
---

---
**Algorithm 2** Planning
---
**Require:** CDiffuser $\psi_\theta(\cdot, \cdot)$, return-to-go predictor $\mathcal{J}_\phi(\cdot, \cdot)$, guidance scale $\rho$, co-variances $\Sigma^i$.
1: $t \leftarrow 1$.
2: **while** not done **do**
3:     Observe state $\boldsymbol{s}_t$; sample $\boldsymbol{\tau}_t^N \sim \mathcal{N}(\boldsymbol{0}, \boldsymbol{I})$
4:     **for** $i = N, N-1, ..., 1$ **do**
5:         Predict return-to-go with $\mathcal{J}_\phi(\hat{\boldsymbol{\tau}}_t^i, i)$.
6:         Sample $\hat{\boldsymbol{\tau}}_t^{i-1}$ using Equation (5).
7:     **end for**
8:     Extract $\hat{\boldsymbol{a}}_t$ form $\hat{\boldsymbol{\tau}}^0$.
9:     Interact with environment using action $\hat{\boldsymbol{a}}_t$.
10:     $t \leftarrow t+1$.
11: **end while**
---

## A.2  IMPACT OF HYPERPARAMETERS ON TRAINING STABILITY.

To evaluate the impact of hyperparameters on training stability, we visualize the training curves of Hopper-Medium with various values of hyperparameters $\xi$, $\zeta$, $\sigma$ and $\lambda_c$, as is shown in Figure 8. It can be concluded that in most of situations, these hyperparameters will not unstable the training process, for example, whatever value $\xi$, $\zeta$, $\sigma$ and $\lambda_c$ take, the training process is stable and CDiffuser converges to a certain point.

## A.3  HYPER-PARAMETERS.

We consider the following hyper-parameter for CDiffuser: Learnign rate, positive bound ($\xi$), negative bound ($\zeta$), fuzzy coefficient ($\sigma$), loss weight of plannign module ($\lambda_d$), loss weight of contrastive learning ($\lambda_c$), loss weight of return predictor $\mathcal{J}_\phi(\cdot, \cdot)$ ($\lambda_v$), guidance scale ($\rho$), diffusion steps ($N$) and the length of subsequent trajectory ($H$). Please notice that both the Planning Module and Ccntrastive Module are trained $1 \times 10^6$ steps, while the return predictor $\mathcal{J}_\phi(\cdot, \cdot)$ is trained $2 \times 10^5$ steps. Detailed hyper-parameter settings for each dataset is provided in  Table 2. Following Diffuser, we perform un-guided sampling for Maze2d environments.

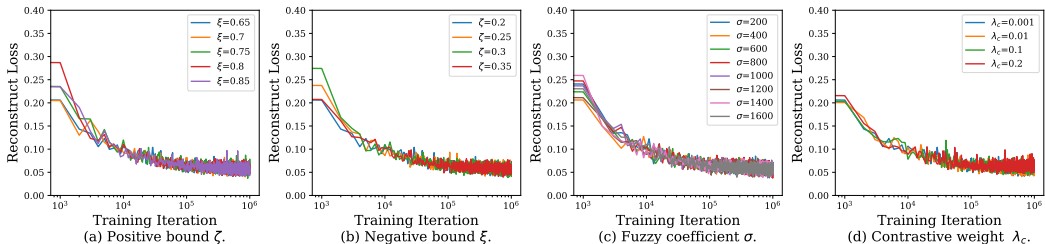

Figure 8: Training curves of hopper-medium-v2 with various values of $\xi$, $\zeta$, $\sigma$ and $\lambda_c$.

Table 2: Hyper-parameter settings for each dataset.

| Dataset | Environment | learning rate | $\xi$ | $\zeta$ | $\sigma$ | $\lambda_d$ | $\lambda_v$ | $\lambda_c$ | $\rho$ | $N$ | $H$ |
|---------|-------------|---------------|-------|---------|----------|-------------|-------------|-------------|--------|-----|-----|
| Med-Expert | HalfCheetah | $2\times10^{-4}$ | 0.65 | 0.05 | $1.6\times10^3$ | 1 | 1 | 0.1 | 0.001 | 20 | 4 |
| Med-Expert | Hopper | $2\times10^{-4}$ | 0.65 | 0.35 | $1.4\times10^3$ | 1 | 1 | 0.001 | 0.0001 | 20 | 32 |
| Med-Expert | Walker2d | $2\times10^{-4}$ | 0.65 | 0.1 | $1\times10^8$ | 1 | 1 | 0.001 | 0.1 | 20 | 32 |
| Medium | HalfCheetah | $2\times10^{-4}$ | 0.85 | 0.2 | $7\times10^2$ | 1 | 1 | 0.01 | 0.001 | 20 | 4 |
| Medium | Hopper | $2\times10^{-4}$ | 0.65 | 0.2 | $8\times10^2$ | 1 | 1 | 0.001 | 0.1 | 20 | 32 |
| Medium | Walker2d | $2\times10^{-4}$ | 0.65 | 0.2 | $4\times10^2$ | 1 | 1 | 0.01 | 0.1 | 20 | 32 |
| Med-Replay | HalfCheetah | $2\times10^{-4}$ | 0.65 | 0.4 | $1\times10^8$ | 1 | 1 | 0.1 | 0.001 | 20 | 4 |
| Med-Replay | Hopper | $2\times10^{-4}$ | 0.55 | 0.2 | $9\times10^2$ | 1 | 1 | 0.001 | 0.1 | 20 | 32 |
| Med-Replay | Walker2d | $2\times10^{-4}$ | 0.6 | 0.05 | $1\times10^8$ | 1 | 1 | 0.1 | 0.1 | 20 | 32 |
| U-Maze | Maze2d | $2\times10^{-4}$ | 5 | 0.2 | $1\times10^8$ | 1 | 1 | 0.1 | - | 20 | 128 |
| Medium | Maze2d | $2\times10^{-4}$ | 0.1 | 0.02 | $1\times10^8$ | 1 | 1 | 0.1 | - | 20 | 256 |
| Large | Maze2d | $2\times10^{-4}$ | 0.6 | 0.01 | $1\times10^8$ | 1 | 1 | 0.1 | - | 20 | 384 |

### A.4 Visualization of positive and negative samples.

We randomly sample a subset of positive samples (states with high returns) and negative samples (states with low returns), as is shown in Figure 9. It can be observed that an agent in a state corresponding to a high return tends to be in a position more conducive to walking or running, such as standing upright; correspondingly, an agent with a state corresponding to a low return will be in a position that is hard to walk, such as having already fallen down or about to fall down. This is reasonable, since poses such as standing upright are more conducive to walking or running, which causes the agent to continue moving and results in a higher return, while poses such as having fallen or about to fall cause the environment to give a stop signal, which results in a lower return.

### A.5 Optimizing $\mathcal{J}\phi(\cdot,\cdot)$ with Equation (14)

Suppose we have the diffuison model $\psi_\theta(\cdot)$ parameterized by $\theta$, and the return predictor $\mathcal{J}_\phi$ parameterized by $\phi$. Following Equation (14), we have

$$\mathcal{L} = \lambda_d\mathcal{L}_d + \lambda_v\mathcal{L}_v + \lambda_c\mathcal{L}_c. \tag{15}$$

Further,

$$\mathcal{L}_d = \mathbb{E}_{\tau_t\in\mathcal{D},t>0,i\sim[1,N]}\left[\|\tau_t - \psi_\theta(\tau_t^i,i)\|^2\right], \tag{16}$$

$$\mathcal{L}_v = \mathbb{E}_{\tau_t\in\mathcal{D},t>0,i\sim[1,N]}[\|\mathcal{J}_\phi(\tau_t^i,i) - v_t\|^2]. \tag{17}$$

The training process can be viewed as a procedure of calculating gradients of all the parameters and updating them, specifically,

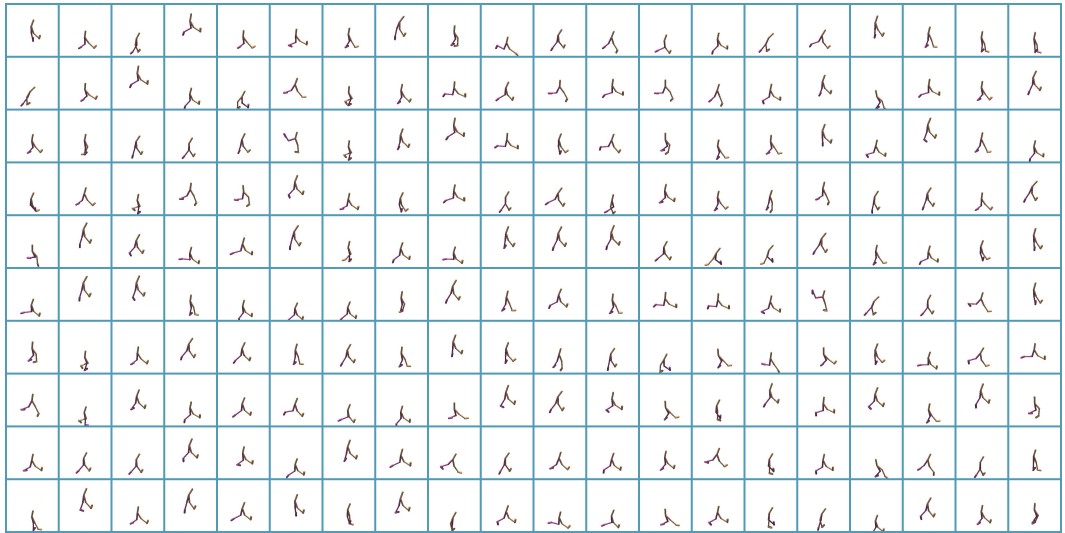

(a) Agents with high-return states.

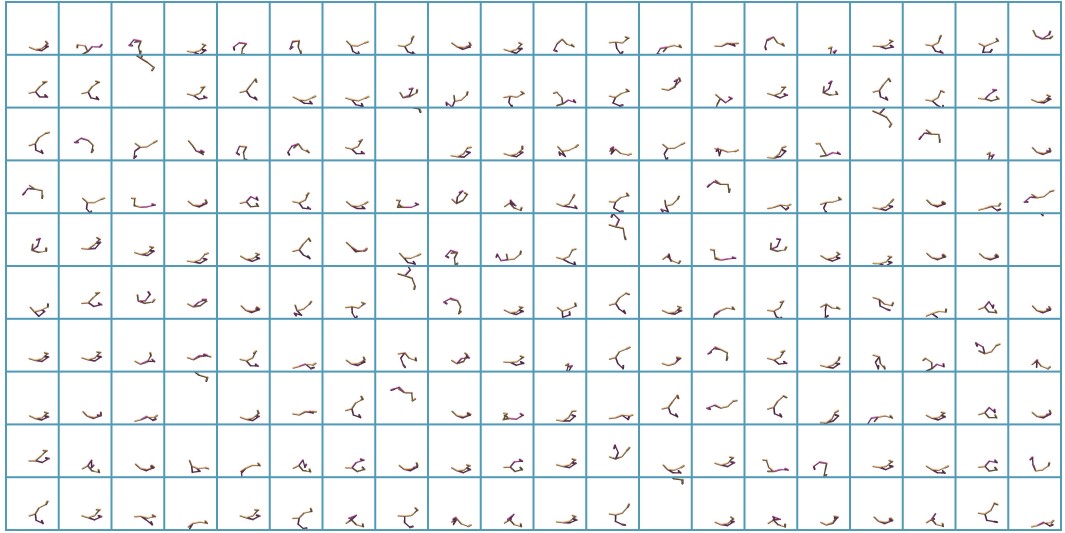

(b) Agents with low-return states.

Figure 9: Visualization of positive samples (states with high returns) and negative samples (states with low returns) in Walker2d-Med-Replay.

$$\nabla\theta = \frac{\partial \mathcal{L}}{\partial \theta} \tag{18}$$

$$= \lambda_d \frac{\partial \mathcal{L}_d}{\partial \theta} + \lambda_v \frac{\partial \mathcal{L}_v}{\partial \theta} + \lambda_c \frac{\partial \mathcal{L}_c}{\partial \theta} \tag{19}$$

$$= \lambda_v \frac{\partial \mathcal{L}_v}{\partial \theta} + \lambda_c \frac{\partial \mathcal{L}_c}{\partial \theta}, \tag{20}$$

$$\nabla\phi = \frac{\partial \mathcal{L}}{\partial \phi} \tag{21}$$

$$= \lambda_d \frac{\partial \mathcal{L}_d}{\partial \phi} + \lambda_v \frac{\partial \mathcal{L}_v}{\partial \phi} + \lambda_c \frac{\partial \mathcal{L}_c}{\partial \phi} \tag{22}$$

$$= \lambda_d \frac{\partial \mathcal{L}_d}{\partial \phi}. \tag{23}$$

Thus, calculating the gradients of $\theta$ with $\mathcal{L}$ is equal to calculate $\theta$ with $\mathcal{L}_d$ and $\mathcal{L}_c$, calculating the gradients of $\phi$ with $\mathcal{L}$ is equal to calculate $\phi$ with $\mathcal{L}_v$, $i.e.$, optimizing the return predictor $\mathcal{J}\phi(\cdot,\cdot)$ with Equation (14) is equal to optimizing it with Equation (12) only.

