# OpenReview forum: "Contrastive Diffuser: Planning Towards High Return States via Contrastive Learning"
_ICLR.cc/2024/Conference — Submitted to ICLR 2024_

### Official Review · Reviewer_ogtz · 2023-10-24

**Soundness:** 2 fair
**Presentation:** 3 good
**Contribution:** 3 good
**Rating:** 5
**Confidence:** 3

**Summary:**

The paper studies the problem of learning state-action trajectory generation with diffusion models. To better leverage the high-return states in the offline dataset, the authors proposed a contrastive learning mechanism to drive the generated trajectory toward the high-return states. Experiments are performed on D4RL benchmarks to validate the idea.

**Strengths:**

The motivation for leveraging contrastive learning to guide the generation process is interesting and reasonable;

The paper is well-written and easily read.

**Weaknesses:**

The method seems on par or slightly worse than the baseline approaches;

The experiments were only conducted on a few simple periodic tasks, which could not sufficiently demonstrate the effectiveness of the method;

There is no analysis of failure modes and limitations.

**Questions:**

What is the task shown in Fig. 5? For visualizing the task, it would be better to align some of the high-return and low-return states to the visual observations of the environment, which may help readers better understand the task.

For Fig. 6, could the authors provide the annotations for the x-axis and y-axis?

It would greatly enhance the paper if the authors could offer a more in-depth analysis of failure cases. Additionally, aside from relatively straightforward periodic tasks, it would be beneficial if the authors explored more complex tasks. Demonstrating the applicability of their approach in scenarios like robot navigation or manipulation would significantly bolster the paper's overall impact and practical relevance.

---

> ### Author Response · Authors · 2023-11-20
> **Responses to Reviewer ogtz**
>
> Thank you very much for your constructive comments! We have conducted more experiments and revised our paper according to your suggestions. The responses to your questions are listed below: \
> **Q1**: The method seems on par or slightly worse than the baseline approaches;\
> **A1**: Thank you for your comment. CDiffuser achieves impressive performance in most cases, although it has slightly worse performance in some cases. Compared with our backbone method Diffuser, CDiffuser demonstrates clear advantages across all the environments, which indicates the effectiveness of contrast in boosting diffusion-based RL methods.
>
>
> **Q2**: The experiments were only conducted on a few simple periodic tasks, which could not sufficiently demonstrate the effectiveness of the method;\
> **A2**: Thank you for your comment.  Following Diffuser, to evaluate the performance of CDiffuser on tasks beyond simple periodic tasks such as hopper, walker2d and halfcheetah, we conduct additional experiments on another three RL environments maze2d-umaze-v1, maze2d-medium-v1, maze2d-large-v1 in Section 4.1. which is not periodic  and tests the model's ability for long-horizon planning. Same to Diffuser, we compare CDiffuser with CQL, IQL and Diffuser. The results are available in Table 1, and are summarized as follows:
> |Environment|CQL|IQL|Diffuser|CDiffuser|
> |:----|:----|:----|:----|:----|
> |Maze2D-UMaze|5.7|47.4|113.9|**142.9±2.2**|
> |Maze2D-Medium|5.0|34.9|121.5|**140.0±0.7**|
> |Maze2D-Large|12.5|58.6|123.0|**131.5±3.2**|
>
> As can be observed, CDiffuser achieves better results than baselines across all three environments, especially on maze2d-umaze-v1, where CDiffuser demonstrated a clear advantage of 25.4% over Diffuser. The results in the Maze2d environments indicate that CDiffuser is also effective in scenarios that require future planning.
>
> **Q3**: There is no analysis of failure modes and limitations.\
> **A3**: Thanks for your suggestion! We have discussed the limitations and future works of CDiffuser in **Conclusion and Discussion**: *the CDiffuser is limited in the case in which a certain state corresponds with both high and low return, as CDiffuser relies on the return on the state to contrast. Nevertheless, the contrast based on the return of state is just the beginning, the contrast on actions also deserves to be explored. We will leave it to future works.*
>
> **Q4**: What is the task shown in Fig. 5? For visualizing the task, it would be better to align some of the high-return and low-return states to the visual observations of the environment, which may help readers better understand the task.\
> **A4**: Thank you for your suggestion. The task shown in Figure 5 is Walker2d-Med-Replay, and we have visualized the high-return and low-return states of the agent in Figure 9 in Appendix A.4.
>
> **Q5**: For Fig. 6, could the authors provide the annotations for the x-axis and y-axis?\
> **A5**: Thanks for you suggestion! We have replotted Figure 6 and provided the annotations for the x-axis and y-axis, please refer to our revised paper for details.
>
> **Q6**: It would greatly enhance the paper if the authors could offer a more in-depth analysis of failure cases. Additionally, aside from relatively straightforward periodic tasks, it would be beneficial if the authors explored more complex tasks. Demonstrating the applicability of their approach in scenarios like robot navigation or manipulation would significantly bolster the paper's overall impact and practical relevance.\
> **A6**: Please refer to **Q3** for analysis of failure cases, **Q2** for more complex tasks like navigation.

---

> > ### Comment · Reviewer_ogtz · 2023-11-22
> >
> > Thanks for the authors’ response, which clarifies the majority of my inquiries.

---

> > > ### Author Response · Authors · 2023-11-22
> > > **Responses to Reviewer ogtz**
> > >
> > > Thank you again for valuable feedback and we would like to request your kindly consideration of improving score or provide us with the reason for why this is not possible.

---

### Official Review · Reviewer_FaMZ · 2023-10-31

**Soundness:** 2 fair
**Presentation:** 3 good
**Contribution:** 2 fair
**Rating:** 3
**Confidence:** 3

**Summary:**

This paper introduces a novel contrastive diffusion probabilistic planning approach to tackle offline reinforcement learning (RL) tasks. It expands upon the foundational Diffuser model, leveraging contrastive learning to enhance the quality of samples by generating high-return trajectories. This focus on sequence modeling within offline RL is both interesting and important. The paper is well-written, though it lacks some specifics, and the visual representations, particularly Figure 1, are insightful.

**Strengths:**

- The focus on sequence modeling in offline-RL is both innovative and significant, addressing a crucial aspect of this field.
- This paper proposed a novel method of integrating contrastive learning that showed improvement over the baseline Diffusers.
- Figure 1 is quite illustrative and intuitive.

**Weaknesses:**

### Missing Details in Methodology:

- The training process of the model remains unclear. While Equation 14 suggests end-to-end training, it is unclear where the contrastive loss is integrated. If added to the diffusion probabilistic model, which aims to reconstruct the un-corrupted trajectories, will this added loss diverge the learning, making the training unstable?
- Is the contrastive loss involved during guidance sampling?

### Insufficient Experiments:

- The claim of 'significant improvements in medium and medium-replay datasets' seems overstated. The improvements are noticeable in only one task from each dataset compared to DD.
- Extending experiments to more complex control tasks or scenarios with high-dimensional state/action spaces would substantiate the method's effectiveness.
- A comparative test incorporating DD + Contrastive Learning would add effectiveness to the proposed method.
- Figure 6 requires more explanation, particularly regarding the methodology for generating and comparing states in each showcased scenario.

### The method introduces several additional hyperparameters, as depicted in Figure 7, indicating a significant sensitivity to these parameters, which could complicate the tuning process.

**Questions:**

- The distinctions among the three models presented in Figure 5 are not very clear to me.
- For the experiments depicted in Figure 6, how are the generated states and actual states obtained in each case?

---

> ### Author Response · Authors · 2023-11-20
> **Responses to Reviewer FaMZ (Q1-Q6)**
>
> Thank you very much for your constructive comments! We have conducted more experiments and revised our paper according to your suggestions. The responses to your questions (Q1-Q6) are listed below: \
> **Q1**: The training process of the model remains unclear. While Equation 14 suggests end-to-end training, it is unclear where the contrastive loss is integrated. If added to the diffusion probabilistic model, which aims to reconstruct the un-corrupted trajectories, will this added loss diverge the learning, making the training unstable?\
> **A1**: The pseudo code of CDiffuser's training process is available in Appendix A.1, and the hyper parameters of  each environment are shown in  Appendix A.3. The contrastive loss is added to the diffusion probabilistic model.  To evaluate how contrastive loss impacts the training process, we visualized the loss during the training process with various hyperparameter values, as is shown in Figure 8 (Appendix A.2). As Figure 8 illustrates,  the added contrastive loss actually has no significant impact on the stability of training.
>
>
> **Q2**：Is the contrastive loss involved during guidance sampling?\
> **A2**：No, the contrastive loss is not involved during guidance sampling.
> The contrastive loss is limited to the training phase. After being guided by the contrastive loss during the training phase, the diffusion probabilistic model performs guided sampling from the distribution of states with higher rewards, leading to better results.
>
> **Q3**：The claim of significant improvements in medium and medium-replay datasets' seems overstated. The improvements are noticeable in only one task from each dataset compared to DD.\
> **A3**：Thank you for your comment. Here, the significant improvements refers to the comparison of CDiffuser and Diffuser, as CDiffuser takes Diffuser as the backbone. As is shown in Table 1, compared with Diffuser, our approach CDiffuser demonstrates clear advantages across all the environments. Nevertheless, we have modified the description of significant improvements in Section 4.2 to avoid any ambiguous expressions.
>
> **Q4**：Extending experiments to more complex control tasks or scenarios with high-dimensional state/action spaces would substantiate the method's effectiveness.\
> **A4**：Thank you for your comment.  Following Diffuser, to evaluate the performance of CDiffuser on tasks beyond simple periodic tasks such as hopper, walker2d and halfcheetah, we conduct additional experiments on another three RL environments maze2d-umaze-v1, maze2d-medium-v1, maze2d-large-v1 in Section 4.1. which is not periodic  and tests the model's ability for long-horizon planning. Same to Diffuser, we compare CDiffuser with CQL, IQL, and Diffuser. The results are available in Table 1, and are summarized as follows:
> |Environment|CQL|IQL|Diffuser|CDiffuser|
> |:----|:----|:----|:----|:----|
> |Maze2D-UMaze|5.7|47.4|113.9|**142.9±2.2**|
> |Maze2D-Medium|5.0|34.9|121.5|**140.0±0.7**|
> |Maze2D-Large|12.5|58.6|123.0|**131.5±3.2**|
>
> As can be observed, CDiffuser achieves better results than baselines across all three environments, especially on maze2d-umaze-v1, where CDiffuser demonstrated a clear advantage of 25.4% over Diffuser. The results in the Maze2d environments indicate that CDiffuser is also effective in scenarios that require future planning.
>
> **Q5**：A comparative test incorporating DD + Contrastive Learning would add effectiveness to the proposed method.\
> **A5**: Thank you for your suggestion! Actually, we once had intentions of applying the contrast mechanism on DD, however, we encountered the same problem with other researchers: we are unable to reproduce the results reported in DD (https://github.com/anuragajay/decision-diffuser/issues/1) . Nevertheless, here are the results of our early experiments:
>
> |Environment|DD|DD + Contrastive Learning|
> |:----|:----|:----|
> |Hopper-Med-Expert|66.7±1.4|**83.0 ± 0.9**|
>
> The results show that incorporating contrastive learning into DD also brings much improvement.
>
> **Q6**：Figure 6 requires more explanation, particularly regarding the methodology for generating and comparing states in each showcased scenario.\
> **A6**: Thank you for your suggestion. We have adjusted Figure 6 and the descriptions corresponding to it in Section 4.4.

---

> ### Author Response · Authors · 2023-11-20
> **Responses to Reviewer FaMZ (Q7 and Q8)**
>
> Thank you very much for your constructive comments! We have conducted more experiments and revised our paper according to your suggestions. The responses to your questions Q7 and Q8 are listed below: \
> **Q7**: The method introduces several additional hyperparameters, as depicted in Figure 7, indicating a significant sensitivity to these parameters, which could complicate the tuning process.\
> **A7**: Thank you for your comment. **Firstly**, every hyperparameter is necessary to perform contrastive learning, since we need them to define the positive/negative indicator, as is introduced in Section 3.2.1.  **Secondly**, Figure 7 shows how the performance of our method varies with the hyperparameter. It can be observed that CDiffuser exhibits a smooth and regular change in performance with hyperparameters various, so it is actually easier for us to tune the parameters.
>
> **Q8**: The distinctions among the three models presented in Figure 5 are not very clear to me.\
> **A8**: Thank you for your suggestion. We have replotted Figure 5. Colored scatters are the states collected during the models' interaction with the environment, while the gray scatters are the states in the dataset.  As can be observed, compared with Decision Diffuser and Diffuser, CDiffuser achieves higher rewards in both in-distribution areas(circled with blue) and out-of-distribution areas(circled with red).

---

> > ### Comment · Reviewer_FaMZ · 2023-11-23
> >
> > I thank the authors for the detailed explanation.
> >
> > **Q1-Q3**: Thank you for your clarification.
> >
> > **Q4**: Maze2D is a simple, low-dimensional state/action space task. The results are not convincing enough to demonstrate the proposed method's effectiveness and robustness.
> >
> > **Q5**: The result on only one task is also not convincing, I can not make any concrete conclusion based on this.
> >
> > **Q6**: Thanks for the revision. It now looks better to me.
> >
> > **Q7**: Tuning so many sensitive hyperparameters will make extending the proposed method to other tasks challenging. The authors only tested their method on Gym-MuJoCo and simple Maze2D tasks. It is still not clear if the proposed method will be effective on other complex tasks, i.e., Kitchen, AntMaze, etc.
> >
> > **Q8**: Thank you! It's much clearer now.

---

> ### Author Response · Authors · 2023-11-23
> **Responses to Reviewer FaMZ (Q1-Q4)**
>
> Thank you for your response and comments! The responses to your questions Q1-Q4 are listed below:\
> **Q1-Q3**: Thank you for your clarification.\
> **A1-A3**: We are pleased that our response has addressed your confusion.
>
> **Q4**: Maze2D is a simple, low-dimensional state/action space task. The results are not convincing enough to demonstrate the proposed method's effectiveness and robustness. \
> **A4**: Thank you for your  comments. We agree that it is worth exploring the performance of CDiffuser in more complex high-dimensional tasks, and we are evaluating CDiffuser on complex tasks  such as Kuka Block Stacking, Antmaze, and the results will be updated soon on our anonymous repo https://anonymous.4open.science/r/ContrastiveDiffuser.
> We believe CDiffuser has great potential to achieve outstanding performance in those tasks for two reasons:  Firstly,  CDiffuser has the potential to gain better performance than Diffuser, which has demonstrated outstanding performance in some high-dimensional tasks like Kuka Block Stacking according to the results of [7,8]. Since the only difference between CDiffuser and Diffuser is the contrastive module, and the effectiveness of contrastive learning in processing high-dimensional information has been proven in the area like CV and RL[1,2,3,4,5,6], CDiffuser is supposed to have outstanding performance on complex tasks with high dimension.
> Secondly, the keys for an RL model to achieve outstanding performance on complex tasks are the state encoder and the policy network: (1) The state encoder is designed to encode the high-dimensional information (i.e., visual information) to a low-dimensional vector, and the state encoders are easily deployed to many RL approaches. Therefore, it is possible to equip CDiffuser with an efficient state encoder;  (2) The policy network is designed to make decisions based on the low-dimensional vectors of states and the rewards. Since CDiffuser's policy has demonstrated excellent performance  on  locomotion tasks and navigation tasks (see Table 1), and CDiffuser can share the same encoder with other methods, it is supposed to achieve outstanding performance in high-dimensional tasks.
>
> [1]Laskin, Michael, Aravind Srinivas, and Pieter Abbeel. "Curl: Contrastive unsupervised representations for reinforcement learning." International Conference on Machine Learning. PMLR, 2020.\
> [2]Ma, Xiao, et al. "Contrastive variational reinforcement learning for complex observations." Conference on Robot Learning. PMLR, 2021.\
> [3] Laskin, Michael, et al. "CIC: Contrastive Intrinsic Control for Unsupervised Skill Discovery." Deep RL Workshop NeurIPS 2021. 2021.\
> [4] Sun, W., Zhang, J., Wang, J., Liu, Z., Zhong, Y., Feng, T., ... & Barnes, N. (2023). Learning Audio-Visual Source Localization via False Negative Aware Contrastive Learning. In Proceedings of the IEEE/CVF Conference on Computer Vision and Pattern Recognition (pp. 6420-6429).\
> [5]Yang, J., Li, C., Zhang, P., Xiao, B., Liu, C., Yuan, L., & Gao, J. (2022). Unified contrastive learning in image-text-label space. In Proceedings of the IEEE/CVF Conference on Computer Vision and Pattern Recognition (pp. 19163-19173).\
> [6] Cherti, M., Beaumont, R., Wightman, R., Wortsman, M., Ilharco, G., Gordon, C., ... & Jitsev, J. (2023). Reproducible scaling laws for contrastive language-image learning. In Proceedings of the IEEE/CVF Conference on Computer Vision and Pattern Recognition (pp. 2818-2829).\
> [7] Ajay, Anurag, et al. "Is conditional generative modeling all you need for decision-making?." arXiv preprint arXiv:2211.15657 (2022).\
> [8] Janner, Michael, et al. "Planning with diffusion for flexible behavior synthesis." arXiv preprint arXiv:2205.09991 (2022).\

---

> ### Author Response · Authors · 2023-11-23
> **Responses to Reviewer FaMZ (Q5-Q8)**
>
> Thank you for your response and comments! The responses to your questions Q5-Q8 are listed below:\
> **Q5**: The result on only one task is also not convincing, I can not make any concrete conclusion based on this.\
> **A5**: Thank you for your  comments. There are two reasons that we did not report the results of  DD + Contrastive Learning in our paper.  Firstly, as we mentioned previously,  we are unable to reproduce the results reported in the paper of DD, although we directly use the public code in the GitHub repo of DD (https://github.com/anuragajay/decision-diffuser). Here are the results.
>
>
> |Environment            |           DD(Reproduced) |  DD(Reported) |
> |  ----  | ----  | ----  |
> |Hopper-Med-Expert      |     66.7±1.4            |     111.8±1.8|
> |Hopper-Med             |          16.7±0.53      |         79.3±3.6|
> |Hopper-Med-Replay       |   25.1±0.8             |    100±0.7|
> |Halfcheetah-Med-Expert  | 39.1±1.3              |    90.6±1.3|
>
> As we can observe,  there are many differences between the results of reproducing and the results reported by DD.
>
> Secondly, we did conduct some experiments on DD + Contrastive Learning, and here are the results.
> |Environment     |                  DD              |        DD + Contrastive Learning|
> |  ----  | ----  | ----  |
> |Hopper-Med-Expert     |    66.7±1.4           |     83.0 ± 0.9|
> |Hopper-Med            |         16.7±0.53      |        26.8 ± 0.8|
>
> As can be observed, DD + Contrastive Learning brings significant improvement (33.4% better than DD on hopper-med-expert, 60.5% better than DD on hopper-med). It can be inferred that Contrastive+DD should exhibit clear improvements on the other tasks.
>
> **Q6**:  Thanks for the revision. It now looks better to me. \
> **A6**:  We are pleased that our response has addressed your confusion.
>
> **Q7**: Tuning so many sensitive hyperparameters will make extending the proposed method to other tasks challenging. The authors only tested their method on Gym-MuJoCo and simple Maze2D tasks. It is still not clear if the proposed method will be effective on other complex tasks, i.e., Kitchen, AntMaze, etc. \
> **A7**: Thanks for your clarification. For the tuning of hyperparameters,  we have offered the suggested hyperparameters in Appendix A. 3. Also, readers can try an easier version of CDiffuser with fewer hyperparameters by directly designating the positive and negative samples to conduct contrast (CDiffuser-Easy). Here is the performance of CDiffuser of the easier version:
>
> |Variants             |       halfcheetah-med-exp |  hopper-med-exp |  walker2d-med-exp |   halfcheetah-med |  hopper-med |  walker2d-med | halfcheetah-med-replay |  hopper-med-replay |  walker2d-med-replay |
> |  ----               |             ----             | ----                  | ----      | ----  |                ----         | ----          | ----                           | ----  |----  |
> |CDiffuser-Easy       |    90.8           |    112.1|  108.2 | 43.8 | 85.4 | 82.9 | 40.0 | 95.4 | 83.2 |
> |CDiffuser            |        92.0   |     112.4| 108.2 | 43.9 | 92.8 | 82.9 | 40.0 | 96.4 | 84.2 |
> |Diffuser            |     88.9   |     103.3 | 106.9 | 42.8 | 74.3 | 79.6 | 37.7  | 93.6 | 70.6 |
>
> The experiment results demonstrate that CDiffuser with fewer hyperparameters also has better performance.
>
> For the evaluation of complex tasks, please refer to **Q4**.
>
> **Q8**: Thank you! It's much clearer now.\
> **A8**: We are pleased that our response has addressed your confusion.

---

> ### Comment · Reviewer_FaMZ · 2023-11-23
>
> **Q5**: The result on only one task is also not convincing, I can not make any concrete conclusion based on this
>
> I happened to run DD (officially released code) on Hopper-Med-Expert from my side; though I couldn't reproduce the reported result, I got a score of around 103 (raw rewards 3345.3, averaged over 10 random samples), much higher than the results obtained from the authors. I encourage authors to communicate with DD's author on the reproduction issues directly and try to seek some help from DD's author.

---

> ### Author Response · Authors · 2023-11-23
> **Responses to Reviewer FaMZ**
>
> Thanks for your suggestion. We conducted experiments by leveraging the same parameters of DD repo (averaged with 10 random seeds). These results of DD and DD + contrastive learning are obtained under the same hyper-parameters and seeds, therefore the results reported above are comparable to some extent. Nevertheless, we will try to communicate with DD's author on the repo's issues to check the training and evaluating settings.

---

### Official Review · Reviewer_F1CC · 2023-10-31

**Soundness:** 1 poor
**Presentation:** 3 good
**Contribution:** 3 good
**Rating:** 3
**Confidence:** 5

**Summary:**

This paper proposes a diffusion-based trajectory generation based on contrasting high-return and low-return training samples, called CDiffuser. The core approach in the method is performing a contrastive learning between generated trajectory and samples in the dataset. The contrastive learning serves as a guidence to the diffusion process, and pushes the generated trajectories towards high return states and away from low return states. Experimental results on Gym show the improvements of the proposed algorithm.

**Strengths:**

- The idea of combining contrastive learning with trajectory generation is somehow novel.

- The analysis on the similary of generated states are informative.

- The ablation study on different loss terms are appreciated. The study on using high-reward samples only is great.

**Weaknesses:**

1. The performance improvement may not be significant. According to Table, 1 the performance is highly comparable to DD.

2. The benchmark only uses Gym.

3. The method is using one step generation from EDP. However, Table 1 and ablation study do not include the comparison against this method.

4. The method highly relies on EDP to make the contrastive loss differentiable through the generated states, from my understanding. However, this could be hard to generalize to other diffusion-based methods.

5. The guidence on return is confusing. The return is predicted from the very **first state** of the **noisy trajectory**, according to the third line after Equation (6). How can the prediction and the learned model be accurate, when solely from a noisy state?  And clarifications on its backpropogation is needed, since it only takes the noisy trajectory input, and the denoising process only takes one step.

6. The original diffuser seperately trains the auxiliary return prediciton model on all data. This modification is not discussed and experimentally validated.

7. Can the authors explain the reasons of intriguing properties presented in 4.4?

8. The contrastive loss Equation (9) seems to not be a common form. Usually the denominator considers all the samples, for example in [2,3]. This is a concern on the correctness of this implementation and a justification is needed.

Based on the points above, I am not convinced the proposed method is sound and could actually work in terms of training.

[1] Bingyi Kang, Xiao Ma, Chao Du, Tianyu Pang, and Shuicheng Yan. Efficient diffusion policies for offline reinforcement learning. arXiv preprint arXiv:2305.20081, 2023.

[2] Oord, Aaron van den, Yazhe Li, and Oriol Vinyals. "Representation learning with contrastive predictive coding." arXiv preprint arXiv:1807.03748 (2018).

[3] Khosla, Prannay, et al. "Supervised contrastive learning." Advances in neural information processing systems 33 (2020): 18661-18673.

**Questions:**

Please see weaknesses.

---

> ### Author Response · Authors · 2023-11-20
> **Responses to Reviewer F1CC (Q1-Q4)**
>
> Thank you very much for your constructive comments! We have conducted more experiments and revised our paper according to your suggestions. The responses to your questions 1-4 are listed below:\
> **Q1**: The performance improvement may not be significant. According to Table 1 the performance is highly comparable to DD.\
> **A1**: Thank you for your comment. Here, the significant improvements refers to the comparison of CDiffuser and Diffuser, as CDiffuser takes Diffuser as the backbone. As is shown in Table 1, compared with Diffuser, our approach CDiffuser demonstrates clear advantages across all the environments. Nevertheless, we have modified the description of significant improvements in Section 4.2 to avoid any ambiguous expressions.
>
> **Q2**: The benchmark only uses Gym. \
> **A2**: Thank you for your comment.  Following Diffuser, to evaluate the performance of CDiffuser on tasks beyond simple periodic tasks such as hopper, walker2d and halfcheetah, we conduct additional experiments on other three RL environments maze2d-umaze-v1, maze2d-medium-v1, maze2d-large-v1 in Section 4.1. which is not periodic and tests the model's ability for long-horizon planning. Same to Diffuser, we compare CDiffuser with CQL, IQL and Diffuser. The results are available at Table 1, and are summarized as follows:
>
> |Environment|CQL|IQL|Diffuser|CDiffuser|
> |:----|:----|:----|:----|:----|
> |Maze2D-UMaze|5.7|47.4|113.9|**142.9±2.2**|
> |Maze2D-Medium|5.0|34.9|121.5|**140.0±0.7**|
> |Maze2D-Large|12.5|58.6|123.0|**131.5±3.2**|
>
>
> We can observe, CDiffuser achieves better results than baselines across all three environments, especially on maze2d-umaze-v1, where CDiffuser demonstrated a clear advantage of 25.4% over Diffuser. The results in the Maze2d environments indicate that CDiffuser is also effective in scenarios that require future planning.
>
>
> **Q3**: The method uses one-step generation from EDP. However, Table 1 and the ablation study do not include the comparison against this method. \
> **A3**: Thank you for your comment. CDiffuser and EDP leverage diffusion models differently. Specifically, following Diffuser and Decision Diffuser, CDiffuser models RL problems as sequence generation problems and leverages diffusion models to generate subsequent  trajectories at each time step. However, methods like EDP and Diffusion-QL leverage diffusion models to act as the policy network, which only predicts action for the current time. In conclusion,  CDiffuser and EDP adopt completely different pipelines.  Considering these, we compare CDiffuser with sequence generation models such as Diffuser and Decision Diffuser rather than EDP.
>
> **Q4**：The method highly relies on EDP to make the contrastive loss differentiable through the generated states, from my understanding. However, this could be hard to generalize to other diffusion-based methods. \
> **A4**:  Thank you for noticing the extension of CDiffuser on other RL methods. We would like to clearify that Firstly, CDiffuser is independent of EDP. As is mentioned in the reply for Q3, CDiffuser and EDP adopted completely different pipelines. Secondly, transplanting the CDiffuser framework to other RL methods is quite convenient. For methods that predict the future states such as Diffuser and Decision Diffuser, the CDiffuser framework can be naturally applied to them as  our contrastive loss $\mathcal{L}_{c}$ in Equation (13) only requires states to contrast. For other methods that only predict actions,  such as Diffusion-QL and EDP, we just need to make little modifications to enable them to predict states while predicting actions, and then our framework can be applied to these methods.  Moreover, the CDiffuser framework is not limited to contrast over states. For example, we can conduct contrast over actions. We leave that as our future work.

---

> ### Author Response · Authors · 2023-11-20
> **Responses to Reviewer F1CC (Q5-Q7)**
>
> Thank you very much for your constructive comments! We have conducted more experiments and revised our paper according to your suggestions. The responses to your questions 5-7 are listed below:\
> **Q5**：The guidance on return is confusing. The return is predicted from the very first state of the noisy trajectory, according to the third line after Equation (6). How can the prediction and the learned model be accurate, when solely from a noisy state? And clarifications on its backpropagation is needed, since it only takes the noisy trajectory input, and the denoising process only takes one step. \
> **A5**: Thank you for your question. Firstly, we would like to clarify that the return is predicted from the whole trajectory, rather than the very first state of it. Secondly, the design of guiding over return is borrowed from Diffuser[1] and Classifier Guided Diffusion[2], in which the return predictor $\mathcal{J}$ is designed to be able to take noisy samples and denoising step as input and output the true returns. Please refer to [1,2] for more details. According to the performance of the Diffuser and CDiffuser, providing guidance by predicting return over noisy trajectory is feasible.
>
> [1] Janner, Michael, et al. "Planning with Diffusion for Flexible Behavior Synthesis." International Conference on Machine Learning. PMLR, 2022. \
> [2] Dhariwal, Prafulla, and Alexander Nichol. "Diffusion models beat gans on image synthesis." Advances in neural information processing systems 34 (2021): 8780-8794.
>
> **Q6**: The original diffuser seperately trains the auxiliary return prediciton model on all data. This modification is not discussed and experimentally validated. \
> **A6**: We separately train the auxiliary return prediction model on all data as is described in Equation (12) and Equation (14), just the same as Diffuser does. The parameters of return predictor $\mathcal{J}$ and diffusion model are independent, therefore updating the parameters of $\mathcal{J}$ separately is equal to updating $\mathcal{J}$'s parameters with $\mathcal{L}$.
>
> **Q7**: Can the authors explain the reasons of the intriguing properties presented in 4.4? \
> **A7**: Thank you for your suggestion. We have discussed the properties of Figure 5 and Figure 6 in Section 4.4. Please refer to our revised paper for details.

---

> ### Author Response · Authors · 2023-11-20
> **Responses to Reviewer F1CC (Q8)**
>
> Thank you very much for your constructive comments! We have conducted more experiments and revised our paper according to your suggestions. The response to your question 8 is listed below:\
> **Q8**: The contrastive loss Equation (9) seems to not be a common form. Usually, the denominator considers all the samples, for example in [2,3]. This is a concern about the correctness of this implementation and a justification is needed. \
> **A8**: Thank you for your question. **Firstly**, sampling a batch as negative samples is commonly adopted in many previous works[1, 4, 5]. The underlying reason is that Equation (9) is iterated for a huge number of steps (1e6 for CDiffuser), which makes Equation (9) equivalent to considering all the samples in the denominator.
> **Secondly**, the contrastive loss in Equation (9) is derived from InfoNCE in [2,3]. We made a few modifications to adapt it to the scenario of a large number of samples. For the negative samples, we randomly sample a batch of size $\kappa$ for calculating the distance of the sample $\hat{s}$ from the original sample space. The reason for this modification is that even if we use the anchor function to classify a small portion of samples as negative (For example, 30%), calculating the distance for all samples is impractical due to the extremely large number of samples (190,000 in the case of med-expert). However, this challenge can be approximately addressed by randomly sampling a batch of size $\kappa$ and iterating a huge number of steps externally (1e6 for CDiffuser), as is the denominator of Equation (9). For the positive samples, we take multiple positive samples for each sample and average the loss values. This extends InfoNCE in [2,3] to the case of multiple positive samples.  The underlying reason is that we want the model learn a distribution close to the distribution of positive samples, not just reproducing a single positive sample. It is worth noting that the numerator of Equation (9) should have a coefficient of $
> \frac{1}{\kappa}$, however, this will transform into a coefficient during the process of gradient descent. Therefore, we merge it into the weight of the contrastive learning loss, $\lambda_c$, and omit it from equation (9).
>
> [1]Wang, Hao, et al. "Knowledge-Adaptive Contrastive Learning for Recommendation." Proceedings of the Sixteenth ACM International Conference on Web Search and Data Mining. 2023.\
> [2] Oord, Aaron van den, Yazhe Li, and Oriol Vinyals. "Representation learning with contrastive predictive coding." arXiv preprint arXiv:1807.03748 (2018).\
> [3] Khosla, Prannay, et al. "Supervised contrastive learning." Advances in neural information processing systems 33 (2020): 18661-18673.\
> [4] Chen, Ting, et al. "A simple framework for contrastive learning of visual representations." International conference on machine learning. PMLR, 2020.\
> [5] Wonsung Lee, Jaeyoon Chun, Youngmin Lee, Kyoungsoo Park, and Sungrae Park. 2022. Contrastive Learning for Knowledge Tracing. In Proceedings of the ACM Web Conference 2022 (WWW '22). Association for Computing Machinery, New York, NY, USA, 2330–2338. https://doi.org/10.1145/3485447.3512105

---

> ### Comment · Reviewer_F1CC · 2023-11-21
>
> Thanks for the response.
>
> **1 & 2:**
>
> Thanks for the clarification and additional experiments. In the revised paper Section 4.4 the authors mentioned Diffuser, DD and proposed method share a similar framework of trajectory generation. So it is a bit contradictory and confusing. Further clarifications are appreciated.
>
> **3 & 4:**
>
> In the submission Section 3.2.2 the authors describe that the proposed method is following Kang et al. 2023 to do one-step denoising. This is not addressed in the response. This paper is using this one-step denoising but not addressing how this one-step denoising affects the performance. Kang et al. definitely focus on a different usage, but the one step-denoising should be well ablated in this submission. The authods did not address the result of removing this one-step denoising.
>
> **5:**
>
> Thanks for the clarification! I think the sentence is already fixed in the revised version.
>
> **6:**
>
> Thanks. I highly suggest that the author could revise the paper to clearly describe it. But this point is still confusing. Diffuser first trains a diffusion model then train J (Page 4, section 3.2), while this paper somehow trains them together according to the response and Equation (14). Could you please further clarify?
>
> **7:**
>
> I appreciate those explanations.
>
> **8:**
>
> Want et al., also consider the positive pairs, i.e. the $\mathbb{N} \cup \\{{v\\}} $ in their equation (8).
> Lee et al. also consider the positive paris in their equations (7) and (8).
> Chen et al. (SimCLR) rely on large batch and do not have explicit negative pairs which is not applicable to CDiffusor. CDiffusor explicitly samples negative samples.
>
> I want to clarify that "all samples" mean both positive and negative pairs in each batch. I understand the difficulty of calculating on all data samples in the dataset. The authors should further clarify this point.
>
> Since the conefficient of contrastive loss is merged into hyper-parameters, and different environment uses different sets of them according to Appendix A.3. The weight of contrastive loss is very small compared to others. The significance of the contrastive loss is questionable.

---

> ### Author Response · Authors · 2023-11-22
> **Responses to Reviewer F1CC (Q1-Q5)**
>
> Thank you for your response and comments! The response to your questions Q1-Q5 are listed below:
>
> **Q1 & 2**:  In the revised paper Section 4.4 the authors mentioned Diffuser, DD and proposed method share a similar framework of trajectory generation. So it is a bit contradictory and confusing. \
> **A1 & 2**: Thank you for your response. The similar framework means that Diffuser, DD and CDiffuser apply diffusion to model the RL as a sequence generation problem. Nevertheless, we have revised our paper to avoid ambiguity.
>
> **Q3 & 4**: In the submission Section 3.2.2 the authors describe that the proposed method is following Kang et al. 2023 to do one-step denoising. This is not addressed in the response. This paper is using this one-step denoising but not addressing how this one-step denoising affects the performance. Kang et al. definitely focus on a different usage, but the one step-denoising should be well ablated in this submission. The authods did not address the result of removing this one-step denoising.\
> **A3 & 4**: Thank you for your clarification! There are three reasons that we did not conduct ablation studies on ont-step denoising. Firstly, we focus on the contrast of states rather than the one-step denoising, and one-step denoising is the contribution of EDP rather than ours. Therefore, we only conduct the ablation studies on contrast. Secondly, multi-step denoising increases the computational requirements significantly, which is beyond the resources we can afford. Thirdly, the ablation studies in EDP demonstrate that one-step generation slightly decreases the performance. Therefore, intuitively speaking, CDiffuser is supposed to achieve better performance with multi-step denoising, which has no negative impact on our current comparison with the baselines.
>
>
> **Q5**: Thanks for the clarification! I think the sentence is already fixed in the revised version.\
> **A5**: We are pleased that our response has addressed your confusion.

---

> ### Author Response · Authors · 2023-11-22
> **Responses to Reviewer F1CC (Q6 and Q7)**
>
> Thank you for your response and comments! The response to your questions Q6 and Q7 are listed below:
>
>
> **Q6**: Thanks. I highly suggest that the author could revise the paper to clearly describe it. But this point is still confusing. Diffuser first trains a diffusion model then train J (Page 4, section 3.2), while this paper somehow trains them together according to the response and Equation (14). Could you please further clarify? \
> **A6**: Of course, we would love to. We have also added the explaination in Appendix A.5 as you suggested. Here are the details of our explaination. Firstly, Diffuser trains $\mathcal{J}$ and diffusion model $\psi$ separately.  Secondly, although the objective of $\mathcal{J}$ and $\psi$ are put into one equation (Equation 14), $\mathcal{J}$ and $\psi$ share no parameters, and their inputs are independent with each other.
>
> Therefore, during the training process, the gradient backpropagation for the parameters of $\mathcal{J}$ and $\psi$ is also independent. The only thing common in $\mathcal{J}$ and $\psi$ is that, they're sharing the same Dataloader, which is actually the same if we set a seed for the training of $\mathcal{J}$ and $\psi$ in Diffuser. In other words, training them 'together' with Equation (14) is equal to training them in the manner of Diffuser.
>
> Here is an example to explain why:\
> Suppose we have the diffuison model $\psi_{\theta}(\cdot)$ parameterized by $\theta$, and the return predictor $\mathcal{J}_{\phi}$ parameterized by $\phi$. At each training iteration, Dataloader provides a batch contains as $\\{s_t, \tau_t, v_t\\}$.
> Following Equation(14),
> $$
> \begin{equation}
>     \mathcal{L} = \lambda_d \mathcal{L_d} + \lambda_v \mathcal{L_v} + \lambda_c \mathcal{L_c}.
> \end{equation}
> $$
> Further,
> $$
> \begin{equation}
>     \mathcal{L}_d = \mathbb{E}\_{{\tau}\_t \in \mathcal{D}, t>0, i \sim [1, N]} [ || {\tau}_t - \psi\_\theta ({\tau}^i_t,i) ||^2 ],
> \end{equation}
> $$
>
> $$
> \begin{equation}
>     \mathcal{L}_v = \mathbb{E}\_{{\tau}\_t\in \mathcal{D}, t>0, i \sim [1, N]}[|| \mathcal{J}\_\phi({\tau}\_t^i, i) - v_t||^2].
> \end{equation}
> $$
>
> The training process can be viewed as a procedure of calculating gradients of all the parameters and updating them, specifically,
> $$
> \begin{align}
>     \nabla\theta
>     &= \frac{\partial \mathcal{L}}{\partial \theta} \\\\
>     &= \lambda_d\frac{\partial \mathcal{L_d}}{\partial \theta} + \lambda_v\frac{\partial \mathcal{L_v}}{\partial \theta} + \lambda_c\frac{\partial \mathcal{L_c}}{\partial \theta} \\\\
>     &= \lambda_d\frac{\partial \mathcal{L_d}}{\partial \theta} + \lambda_c\frac{\partial \mathcal{L_c}}{\partial \theta},
> \end{align}
> $$
> $$
> \begin{align}
>     \nabla\phi
>     &= \frac{\partial \mathcal{L}}{\partial \phi} \\\\
>     &= \lambda_d\frac{\partial \mathcal{L_d}}{\partial \phi} + \lambda_v\frac{\partial \mathcal{L_v}}{\partial \phi} + \lambda_c\frac{\partial \mathcal{L_c}}{\partial \phi}  \\\\
>     &= \lambda_v\frac{\partial \mathcal{L_v}}{\partial \phi}.
> \end{align}
> $$
>
> Thus, calculating the gradients of $\theta$ with $\mathcal{L}$ is equal to calculating $\theta$ with $\mathcal{L}_d$ and $\mathcal{L}_c$, calculating the gradients of $\phi$ with $\mathcal{L}$ is equal to calculating $\phi$ with $\mathcal{L}_v$, $i.e.$,  training them 'together' with Equation (14) is equal to training them in the manner of Diffuser.
>
> **Q7**: I appreciate those explanations. \
> **A7**: We are pleased that our response has addressed your confusion.

---

> ### Author Response · Authors · 2023-11-22
> **Responses to Reviewer F1CC (Q8)**
>
> Thank you for your response and comments! The response to your question Q8 are listed below:
>
> **Q8(1)**: Want et al., also consider the positive pairs, i.e. the $\mathbb{N} \cup \{{v\}} $ in their equation (8). Lee et al. also consider the positive paris in their equations (7) and (8). Chen et al. (SimCLR) rely on large batch and do not have explicit negative pairs which is not applicable to CDiffusor. CDiffusor explicitly samples negative samples.I want to clarify that "all samples" mean both positive and negative pairs in each batch. I understand the difficulty of calculating on all data samples in the dataset. The authors should further clarify this point. \
> **A8(1)**: Thank you for your comment. Firstly, different from the works mentiond in the references[1,2,3] which consider one positive sample for each sample, we consider $\kappa$ positive samples to guide the learned distribution. Inspired by [4,5], we adopt Equation (9) as the learning target to pull the states in the generated trajectory toward the high-return states and away from the low-return states.
> Secondly, Equation (9) is similar to the objectives in [1,2,3]. Most of the works mentioned above is based on the InfoNCE:
> $$
> \begin{align}
>     \theta^*_{InfoNCE}
>     & = \underset{\theta}{\arg \min} -\mathbb{E} \left[  log\frac{exp(sim(f(s),s^+)/T)} {exp(sim(f(s),s^+)/T) + \sum_{s^- \in S^-} exp(sim(f(s),s^-)/T) }   \right] \\\\
>     & = \underset{\theta}{\arg \min} \mathbb{E} \left[  log\frac{exp(sim(f(s),s^+)/T) + \sum_{s^- \in S^-} exp(sim(f(s),s^-)/T) }{exp(sim(f(s),s^+)/T)}  \right] \\\\
>     & = \underset{\theta}{\arg \min} \mathbb{E}  \left[  log(1+ \frac{\sum_{s^- \in S^-} exp(sim(f(s),s^-)/T)}{exp(sim(f(s),s^+)/T)} )        \right],
> \end{align}
> $$
> and optimizing this equation is similar to optimizing (though the gradients are different):
> $$
> \begin{align}
>     & \underset{\theta}{\arg \min} \mathbb{E}  \left[  log(\frac{\sum_{s^- \in S^-} exp(sim(f(s),s^-)/T)}{exp(sim(f(s),s^+)/T)} )        \right] \\\\
>     &= \underset{\theta}{\arg \min} - \mathbb{E}  \left[  log(\frac{exp(sim(f(s),s^+)/T)}{\sum_{s^- \in S^-} exp(sim(f(s),s^-)/T)} )        \right].
> \end{align}
> $$
> Thus, Equation (9) is similar to the InfoNCE mentioned in [1,2,3].
>
> [1]Wang, Hao, et al. "Knowledge-Adaptive Contrastive Learning for Recommendation." Proceedings of the Sixteenth ACM International Conference on Web Search and Data Mining. 2023.\
> [2] Oord, Aaron van den, Yazhe Li, and Oriol Vinyals. "Representation learning with contrastive predictive coding." arXiv preprint arXiv:1807.03748 (2018).\
> [3] Khosla, Prannay, et al. "Supervised contrastive learning." Advances in neural information processing systems 33 (2020): 18661-18673.\
> [4] Florian Schroff, Dmitry Kalenichenko, and James Philbin. Facenet: A unified embedding for face recognition and clustering. In Proceedings of the IEEE conference on computer vision and pattern recognition, pp. 815–823, 2015\
> [5]Kihyuk Sohn. Improved deep metric learning with multi-class n-pair loss objective. Advances in neural information processing systems, 29, 2016.
>
> **Q8(2)**: Since the conefficient of contrastive loss is merged into hyper-parameters, and different environment uses different sets of them according to Appendix A.3. The weight of contrastive loss is very small compared to others. The significance of the contrastive loss is questionable. \
> **A8(2)**: Thank you for your comment. Firstly, the weight of contrastive loss is used to balance the objectives  in $\mathcal{L}$, and the values of each weight in Equation (14) is adjusted based on the performance.  Secondly, the expiremental results demonstrate the effectiveness of contrastive loss: **(1)** The hyper-parameters analysis in Figure 7 (d) show that, the weight of contrastive loss has significant impact on the performance, although its value is small. **(2)** The ablation study demonstrates that removing the contrastive loss (Diffuser-C) decreases the performance on all of the 9 locomotion tasks, which shows the significance of the contrastive loss.

---

### Official Review · Reviewer_VS91 · 2023-10-31

**Soundness:** 4 excellent
**Presentation:** 4 excellent
**Contribution:** 4 excellent
**Rating:** 10
**Confidence:** 3

**Summary:**

This paper combines a trajectory planning method based on a diffusion model and contrastive learning to select states with higher returns. States are grouped into fuzzy sets of low and high reward and then used to constrain the trajectory planning by pulling states towards regions of higher return. An extensive ablation study, comparison with state of the art and hyperparameter search shows good results and the importance of all components.

**Strengths:**

The method is suprisingly simple yet effective. It can be probably easily adopted for a wide range planning problems or even tasks without explicit trajectories beyond the typical RL tasks in table 1.

**Weaknesses:**

It is hard to find weaknesses in this paper. Sometimes the sentences are a bit long and convey a lot of concepts at the same time which is not necessarily bad but harder to understand. One example is the sentence around equation 13. The impact of predictions in the future of planning could be elaborated a bit more, but this is just an example to illustrate my point.

While being best and second best in the med-replay datasets, it could be argued if being so close to the other results can be called significant improvements and highlighting the second best is potentially done to have the results in the best possible light. However, the authors put their results in ample perspective and give reasonable hypothesis about the impact of expert examples.

**Questions:**

- More a suggestion, Figure captions like Figure 2 could provide more information. The general function of both modules as take away message for the reader could improve the figure understanding even though it appears in the text pointing to this figure

- In figure 5 I find it hard to see what is supposed to be in and out of distribution. Maybe some circles could help making the points from section 4.4. All three figures also look very alike. I get the idea of comparison here but not sure about the overall value of this. The nuanced color changes are also hard to see and some people are color blind.

---

> ### Author Response · Authors · 2023-11-20
> **Responses to Reviewer VS91**
>
> Thank you very much for your admiration of our work!  As you commented, CDiffuser can be further applied in many other RL tasks. Moreover, CDiffuser is not limited to performing contrastive learning over states but also can be extended to perform contrastive learning over actions. Also, thank you for your constructive comments! We have made some changes to the paper based on your suggestions. The responses to your questions and comments are listed below:
>
> **Q1**: While being the best and second best in the med-replay datasets, it could be argued if being so close to the other results can be called significant improvements, and highlighting the second best is potentially done to have the results in the best possible light.\
> **A1**: Thank you for your comment. Here, the significant improvements refers to the comparison of CDiffuser and Diffuser, as CDiffuser takes Diffuser as the backbone. As is shown in Table 1, compared with Diffuser, our approach CDiffuser demonstrates clear advantages across all the 12 settings. Nevertheless, we have modified the description of significant improvements in Section 4.2 to avoid any ambiguous expressions.
>
> **Q2**: More a suggestion, Figure captions like Figure 2 could provide more information. \
> **A2**: Thank you for your suggestions. We have added a brief description of the pipeline in the caption of Figure 2 to improve the figure understanding.
>
> **Q3**: In figure 5 I find it hard to see what is supposed to be in and out of distribution. Maybe some circles could help making the points from section 4.4. All three figures also look very alike. I get the idea of comparison here but not sure about the overall value of this. The nuanced color changes are also hard to see and some people are color blind. \
> **A3**: Thank you for your suggestion. We have replotted Figure 5. Colored scatters are the states collected during the models' interaction with the environment, while the gray scatters are the states in the dataset.  As can be observed, compared with Decision Diffuser and Diffuser, CDiffuser achieves higher rewards in both in-distribution areas(circled with blue) and out-of-distribution areas(circled with red).

---

> > ### Comment · Reviewer_VS91 · 2023-11-22
> >
> > Thank you for the clarifications. I have read all the discussion of all other reviewers. From the perspective of trajectory generation and contrastive learning I can see the value of this approach and would be happy if it were to be discussed in the larger research community at the conference. I think the method is interesting for application in my domain and the results show decently honest that it is needed to further evaluate and improve it because it does not beat the state of the art by a large margin. I still see the large methodical contribution of the method. However, I understand the critique of my dear colleagues from the perspective of generalization to other diffusion based methods and while I still think this paper warrants acceptance I would not be upset if I were to be overruled.

---

> > > ### Author Response · Authors · 2023-11-22
> > > **Responses to Reviewer VS91**
> > >
> > > we are deeply grateful for your admiration of our work!  We will continue to work hard to make more contributions.

---

### Author Response · Authors · 2023-11-20
**Response to all the reviewers and area chair**

We would like to express our sincere appreciation to all reviewers for your constructive feedback. We have revised our paper according to the reviewers' feedbacks (highlighted in blue in the paper). Specifically, we have made the following changes:
1. we conduct additional experiments on three RL environments maze2d-umaze-v1, maze2d-medium-v1, and maze2d-large-v1 in Section 4. Following Diffuser, we compare CDiffuser with CQL, IQL, and Diffuser. The results are illustrated in Table 1.
2. We modify the descriptions in the paper to avoid ambiguity, for example, the description of significant.
3. For Figure 5, we have  adjusted it to distinguish the high-reward states and low-reward states, and have updated the discussions about Figure 5 in  Section 4.4. Moreover, we have visualized the agent with high-return and low-return states in Figure 5 in the Appendix A.4.
4. For Figure 6, we have replotted it and made some adjustments for better understanding, e.g., we provided the annotations for the x-axis and y-axis of it. We have also updated the discussions of Figure 6. Please refer to Figure 6 and Section 4.4 for more details.
 5. We have added additional descriptions about the pipeline in the caption of Figure 2 to improve the figure understanding.
6. We have discussed the limitations and future works of CDiffuser in the section of Conclusion and Discussion.

---

### Meta-Review · Area_Chair_kwxv · 2023-12-05

**Metareview:**

Summary: The paper introduces a novel contrastive diffusion probabilistic planning approach for offline reinforcement learning (RL) tasks, building upon the foundational Diffuser model. The addition of contrastive learning enhances the quality of samples, generating high-return trajectories.

Strengths: The paper is generally well-written and the focus on sequence modeling in offline RL is timely.

Weaknesses: The methodology lacks clarity, especially in the training process, and there are concerns about the stability of training with added contrastive loss. The experiments are criticized for lacking depth and not exploring more complex tasks or scenarios. Specific figures, such as Figure 6, need more explanation, and the introduction of additional hyperparameters raises concerns about tuning complexity. Results are incomplete - baselines do not have variance reported (while CDiffuser does), making it impossible to judge significance of the results. I also am concerned that there is little justification for the contrastive objective "pulling" the diffusion model towards high-return states. Figure 1 provides some intuition, but is not strong enough to act as justification.

Suggestions: First, I want to commend the authors on a heroic job on their rebuttal. The additional results, clarification, and engagement with reviewers is a lot of work and I want to recognize that. However, while reviewers engaged with the authors, they did not raise scores. I am also still not comfortable with the justification of the method and how results are reported. I suggest addressing these in the next revision of your paper.

**Justification For Why Not Higher Score:**

Results are incomplete - baselines do not have variance reported (while CDiffuser does), making it impossible to judge significance of the results. I also am concerned that there is little justification for the contrastive objective "pulling" the diffusion model towards high-return states. Figure 1 provides some intuition, but is not strong enough to act as justification.

**Justification For Why Not Lower Score:**

N/A

---

### Decision · Program_Chairs · 2024-01-16

Reject